# Rifampicin exposure reveals within-host *Mycobacterium tuberculosis* diversity in patients with delayed culture conversion

**Charlotte Genestet**[1,2]*, **Elisabeth Hodille**[1,2], **Alexia Barbry**[1,2], **Jean-Luc Berland**[1,3], **Jonathan Hoffmann**[1,3], **Emilie Westeel**[1,3], **Fabiola Bastian**[4], **Michel Guichardant**[5], **Samuel Venner**[6], **Gérard Lina**[1,2,7], **Christophe Ginevra**[1,2], **Florence Ader**[1,8], **Sylvain Goutelle**[6,7,9], **Oana Dumitrescu**[1,2,7]

**1** CIRI—Centre International de Recherche en Infectiologie, Ecole Normale Supérieure de Lyon, Université Claude Bernard Lyon-1, Inserm U1111, CNRS UMR5308, Lyon, France, **2** Hospices Civils de Lyon, Institut des Agents Infectieux, Laboratoire de bactériologie, Lyon, France, **3** Fondation Mérieux, Emerging Pathogens Laboratory, Lyon, France, **4** Plateforme DTAMB, CNRS, Université Lyon 1, Villeurbanne, France, **5** CarMeN laboratory, INSA Lyon, INSERM U1060, INRA U1397, Université Lyon 1, Villeurbanne, France, **6** Laboratoire de Biométrie et Biologie Évolutive, CNRS UMR 5558, Université Lyon 1, Villeurbanne, France, **7** Université Lyon 1, Facultés de Médecine et de Pharmacie de Lyon, Lyon, France, **8** Hospices Civils de Lyon, Service des Maladies infectieuses et tropicales, Lyon, France, **9** Hospices Civils de Lyon, Groupement Hospitalier Nord, Service pharmaceutique, Lyon, France

* charlotte.genestet@gmail.com

**Data Availability Statement:** Sequences have been submitted to European Nucleotide Archive (ENA) under accession number PRJEB37306.

## Abstract

*Mycobacterium tuberculosis* (Mtb) genetic micro-diversity in clinical isolates may underline mycobacterial adaptation to tuberculosis (TB) infection and provide insights to anti-TB treatment response and emergence of resistance. Herein we followed within-host evolution of Mtb clinical isolates in two cohorts of TB patients, either with delayed Mtb culture conversion (> 2 months), or with fast culture conversion (< 2 months). We captured the genetic diversity of Mtb isolates obtained in each patient, by focusing on minor variants detected as unfixed single nucleotide polymorphisms (SNPs). To unmask antibiotic tolerant sub-populations, we exposed these isolates to rifampicin (RIF) prior to whole genome sequencing (WGS) analysis. Thanks to WGS, we detected at least 1 unfixed SNP within the Mtb isolates for 9/15 patients with delayed culture conversion, and non-synonymous (ns) SNPs for 8/15 patients. Furthermore, RIF exposure revealed 9 additional unfixed nsSNP from 6/15 isolates unlinked to drug resistance. By contrast, in the fast culture conversion cohort, RIF exposure only revealed 2 unfixed nsSNP from 2/20 patients. To better understand the dynamics of Mtb micro-diversity, we investigated the variant composition of a persistent Mtb clinical isolate before and after controlled stress experiments mimicking the course of TB disease. A minor variant, featuring a particular mycocerosates profile, became enriched during both RIF exposure and macrophage infection. The variant was associated with drug tolerance and intracellular persistence, consistent with the pharmacological modeling predicting increased risk of treatment failure. A thorough study of such variants not necessarily linked to canonical drug-resistance, but which are prone to promote anti-TB drug tolerance, may be crucial to prevent the subsequent emergence of resistance. Taken together, the present findings support the further exploration of Mtb micro-diversity as a promising tool to detect patients at

**Funding:** This work was supported by the LABEX ECOFECT (ANR-11-LABX-0048) of Université de Lyon, within the program "Investissements d'Avenir" (ANR-11-IDEX-0007) operated by the French national research agency (Agence nationale de la recherché, ANR). CGe was the recipient of this grant. The funders had no role in study design, data collection and analysis, decision to publish, or preparation of the manuscript.

**Competing interests:** The authors have declared that no competing interests exist.

risk of poorly responding to anti-TB treatment, ultimately allowing improved and personalized TB management.

## Author summary

Tuberculosis (TB) is caused by *Mycobacterium tuberculosis* (Mtb), bacteria that are able to persist inside the patient for many months or years, thus requiring long antibiotic treatments. Here we focused on TB patients with delayed response to treatment and we performed genetic characterization of Mtb isolates to search for sub-populations that may tolerate anti-TB drugs. We found that Mtb cultured from 9/15 patients contained different sub-populations, and *in vitro* drug exposure revealed Mtb sub-populations in 6/15 isolates, none related to known drug-resistance mechanisms. By contrast, drug exposure revealed Mtb sup-populations in 2/20 isolates in the control cohort of patients with fast culture conversion. Furthermore, we characterized a Mtb variant isolated from a sub-population growing in the presence of rifampicin (RIF), a major anti-TB drug. We found that this variant featured a modified lipidic envelope, and that it was able to develop in the presence of RIF and inside human macrophage cells. We performed pharmacological modelling and found that this kind of variant may be related to a poor response to treatment. In conclusion, searching for particular Mtb sub-populations may help to detect patients at risk of treatment failure and provide additional guidance for TB management.

## Introduction

Tuberculosis (TB) caused by the *Mycobacterium tuberculosis* (Mtb) complex remains one of the most prevalent and deadly infectious diseases; it was responsible for 10 million cases and 1.45 million deaths worldwide in 2018 [1]. One of the most remarkable features of Mtb infection is its chronicity, with long periods of latency, linked to the ability of the tubercle bacilli to persist in the host tissues. TB disease therefore requires a long duration of antibiotic treatment to achieve sterilization of both multiplying and dormant bacilli. Anti-TB drug resistance detection is strongly recommended upon TB diagnosis, and mandatory in some high-income countries, as drug resistance is known to hamper treatment efficacy of first-line anti-TB drugs [2,3]. However, persistent infections with delayed response to treatment may be observed without any *in vitro* proven antibiotic resistance. Hypothetically, pre-existing sub-populations enclosed within Mtb clinical isolates that are antibiotic tolerant could be responsible for such persistent infections [4,5], but evidence of this is still lacking.

 Since the introduction of next generation sequencing (NGS), whole genome sequencing (WGS) analysis performed on clinical Mtb isolates allowed the genetic particularities of Mtb sub-populations or variants to be revealed. Minor variants, stemmed from the initial infecting strain (hereafter referred to as the micro-diversity phenomenon), are therefore frequently present within the Mtb population before treatment onset and may also be revealed during TB treatment [6–12]. Some of these variants harbor drug resistance mutations, whilst others carry SNP in loci involved in modulation of innate immunity and in the production of Mtb cell envelop lipids [12–16]. Although the emergence of variants occurring in loci not directly related to drug-resistance is still poorly understood [17–19], this could be the key to understanding the mechanisms involved in Mtb host-adaptation possibly leading to treatment failure [20,21].

This raises the question of the evolving patterns of within-host Mtb micro-diversity in response to anti-TB treatment, and its possible link with patient outcome, beyond the canonical Mtb drug resistance phenomenon.

Delayed culture conversion (after 2 months of treatment) has been considered to be a predictor of treatment failure in TB patients [22]. We therefore focused on Mtb isolated from patients managed for drug-susceptible TB in our center between 2014 and 2017, without culture conversion after 2 months of well conducted anti-TB treatment. We captured the genetic diversity of both the initial and the last Mtb isolate obtained in each patient and we hypothesized that Mtb micro-diversity may be involved in bacilli persistence and in drug tolerance. Therefore, to unmask antibiotic tolerant sub-populations, we exposed the last Mtb isolate to rifampicin (RIF), a major anti-TB drug, prior to WGS analysis. To better understand the dynamics of Mtb micro-diversity during the course of TB, we investigated the variant composition of a persistent Mtb clinical isolate before and after controlled stress experiments chosen to mimic the course of TB disease. Importantly, a minor Mtb variant revealed after RIF exposure was found to be associated with both drug-tolerance and intra-macrophage persistence, suggesting that Mtb micro-diversity should thus be further considered to improve TB management and prevent treatment failure.

## Results

### Mtb minor variants are detected during the course of TB and upon RIF exposure

From 435 susceptible TB patients managed in our center between 2014 and 2017, 20 presented positive sputum culture later than 2 months after the beginning of treatment, without culture conversion between time points (*i.e.* 15 days and 1 month after treatment initiation). Five patients were lost to follow-up; 15 patients were therefore enrolled in the study (Fig 1). Thirteen patients presented pulmonary TB, in 2 patients TB was disseminated; all patients were smear positive. Clinical severity scores were high (Bandim TB score > 4 or Malnutrition Universal Screening Tool (MUST) > 3) in all patients. Three patients died, the other 12 patients presented no relapse at 2 years follow-up; all-but-two patients received prolonged anti-TB treatment (Table 1). We also enrolled 20 patients with smear-positive TB without delayed in culture conversion who composed the control cohort (Table 2). Among these patients, 16 had pulmonary TB and 4 had disseminated TB. Clinical severity scores were high for 11/20 (55%) patients. We observed no casualties and no relapse after 6 to 12 months of anti-TB treatment.

For each patient, genomes of both early and last Mtb isolates (for the delayed culture conversion cohort only) were analyzed with a special focus on minor variants, meaning unfixed mutations. Moreover, to unmask RIF-tolerant sub-populations, the last isolates for the delayed culture conversion cohort or the initial isolates for control cohort, were exposed *in vitro* to RIF at 1x minimum inhibitory concentration (MIC) for in Mycobacterial Growth Indicator Tube (MGIT) 4 weeks prior to WGS. Regarding initial isolates of the control cohort, 4/20 (20%) presented at least 1 unfixed mutation, among which 2 non-synonymous single nucleotide polymorphisms (nsSNP), and 2 additional nsSNP from 2/20 (10%) isolates were revealed by RIF exposure (Table 2). Among the initial isolates of the delayed culture conversion cohort, 7/15 (47%) presented at least 1 unfixed mutation according to WGS, among which 4 nsSNP, while among the last isolates 6 additional nsSNP from 5 isolates were detected, and one previously unfixed nsSNP became fixed. Interestingly, among the 9 unfixed nsSNP revealed by RIF exposure in 6/15 (40%) isolates, all were detected at low frequency thanks to targeted NGS in the last isolate and none were linked to drug resistance (Table 1). Moreover, the bacterial growth was explored during subsequent exposure to 4xMIC RIF, by the time to positivity (TTP) approach in MGIT which reflects the CFU count, a low TTP indicating a high bacterial load

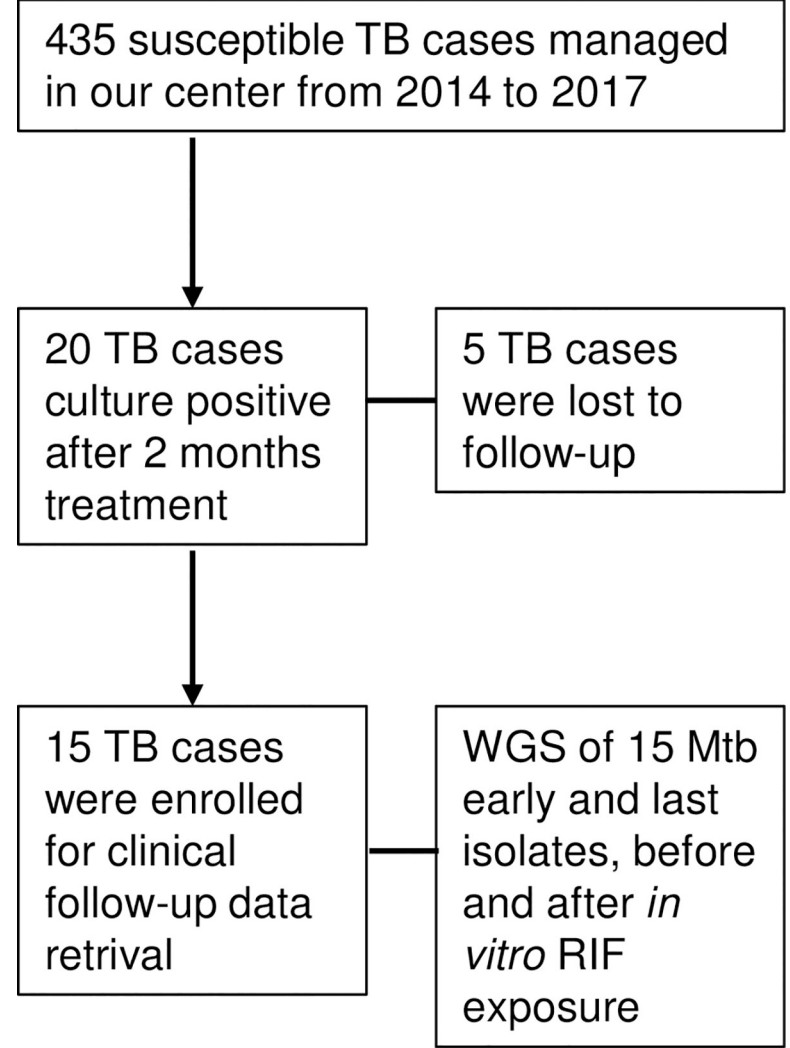

**Fig 1. Flowchart of delayed conversion TB patients enrollment for WGS of early and last Mtb isolates, before and after *in vitro* rifampicin (RIF) exposure.**

and vice versa [23,24]. Growth controls without exposure to RIF did not reveal significant difference in fitness in the absence of antibiotic pressure. Conversely, 10/15 patient isolates tested had lower TTP during subsequent exposure to 4xMIC RIF for isolates previously exposed to RIF compared to paired initial and/or last isolates, revealing a better fitness upon subsequent drug exposure. TTP were particularly shortened for isolates previously exposed to RIF of patients P1 (8.8 days), P4 (7.1 days), P6 (7.3 days), and P10 (7.4 days) compared to the paired last isolate (S1 Fig). We hypothesized that isolates previously exposed to RIF and showing shortened TTP upon subsequent RIF incubation, may harbor antibiotic-tolerant variants, which was then further explored.

## A minor variant of Mtb clinical isolate is selected in response to both rifampicin exposure and macrophage infection

Given the shortening of growth delay upon RIF exposure (as assessed by the TTP with the MGIT-BACTEC system) and the non-synonymous nature of the non-fixed variant (C

**Table 1. Mtb variants identified emerging from persistent clinical isolates and after 4 weeks of exposure to 1x MIC RIF *in vitro*.**

| Patient/ Mtb lineage | Clinical presentation/ smear | Comorbidity | Duration of positive culture, months | Position | Gene | Mutation type | Variant frequencies (%) | | | Functional categories | Severity score Bandim score/ MUST | Outcome (2 years follow-up) |
|---|---|---|---|---|---|---|---|---|---|---|---|---|
| | | | | | | | Initial sample | Last sample | *in vitro* RIF exposure | | | |
| P1/L2 | Disseminated TB/AFB + | Immuno-suppressive therapy/ Diabetes | 2 | 3280555 | mas Rv2940c | nsSNP | ND | **2.3**\* | 44 | LM | 6/6 | No relapse after 18 months anti-TB treatment |
| P2/L2 | Pulmonary TB/AFB + | None | 6 | 2892553 2592179 | Rv2568c Rv2319c | nsSNP nsSNP | **0.10**\* 30 | **9.5**\* 43 | 22 52 | CH VDA | 5/3 | No relapse after 10 months anti-TB treatment |
| P3/L2 | Disseminated TB/AFB + | HIV | 3 | 15876 2609982 2199585 899963 | pknB Rv0014c cysE Rv2335 Rv1949c cpsY Rv0806c | nsSNP nsSNP sSNP nsSNP | ND ND 36 ND | **0.14**\* **0.11**\* 100 42 | 36 62 100 ND | RP IMR CH CWCP | 7/4 | No relapse after 12 months anti-TB treatment |
| P4/L3 | Pulmonary TB/AFB + | Hepatitis C | 4 | 555603 1803827 | Rv0465c hisA Rv1603 | nsSNP sSNP | ND ND | 22 **13.5**\* | **0.20**\* 97 | IMR IMR | 8/4 | Death after 18 months anti-TB treatment |
| P5/L4 | Pulmonary TB/ AFB + | Immuno-suppressive therapy/ Diabetes | 2 | No variant identified | N/A | N/A | N/A | N/A | N/A | N/A | 7/4 | No relapse after 9 months anti-TB treatment |
| P6/L4 | Pulmonary TB/AFB + | Diabetes | 2 | 328608 600809 4327302 | Rv0272c hemA Rv0509 ethA Rv3854c | sSNP sSNP DEL | ND ND ND | 100 100 100 | 100 100 100 | CH IMR IMR | 5/2 | Death by secondary *Aspergillus* infection |
| P7/L4 | Pulmonary TB/AFB + | Hepatitis C | 3 | 4177494 | Rv3728 | sSNP | 20 | 93 | 45 | CWCP | 9/6 | No relapse after 9 months anti-TB treatment |
| P8/L4 | Pulmonary TB/AFB + | None | 2 | 3040405 | miaA Rv2727c | nsSNP | ND | **0.8**\* | 20 | IMR | 6/6 | Death before treatment completion |
| P9/L4 | Pulmonary TB/AFB + | None | 2 | 115217 966395 | nrp Rv0101 moaA Rv0869c | sSNP nsSNP | 25 22 | **4.8**\* **11.8**\* | ND ND | LM IMR | 2/5 | No relapse after 9 months anti-TB treatment |
| P10/L4 | Pulmonary TB/AFB + | None | 2 | 853811 3771284 1150503 | phoR Rv0758 relK Rv3358 kdpD Rv1028c | sSNP sSNP nsSNP | ND ND ND | 35 29 56 | 20 18 **15.7**\* | RP VDA RP | 5/4 | No relapse after 6 months anti-TB treatment |
| P11/L4 | Pulmonary TB/AFB + | None | 2 | 183015 379185 | fadE2 Rv0154c Rv0311 | sSNP nsSNP | 15 17 | 100 100 | 100 100 | LM CH | 7/4 | No relapse after 9 months anti-TB treatment |
| P12/L4 | Pulmonary TB/AFB + | None | 3 | 451849 1689565 3513662 | Rv0374c Rv1498c nuoD Rv3148 | nsSNP nsSNP nsSNP | ND ND **0.1**\* | **0.1**\* **1.1**\* **2.7**\* | 36 21 20 | IMR IMR IMR | 5/4 | No relapse after 9 months anti-TB treatment |
| P13/L4 | Pulmonary TB/AFB + | None | 2 | 1369286 | Rv1226c | nsSNP | ND | **9.1**\* | 40 | CWCP | 3/4 | No relapse after 6 months anti-TB treatment |

*(Continued)*

**Table 1.** (Continued)

| Patient/ Mtb lineage | Clinical presentation/ smear | Comorbidity | Duration of positive culture, months | Position | Gene | Mutation type | Variant frequencies (%) | | | Functional categories | Severity score Bandim score/ MUST | Outcome (2 years follow-up) |
|---|---|---|---|---|---|---|---|---|---|---|---|---|
| | | | | | | | Initial sample | Last sample | *in vitro* RIF exposure | | | |
| P14/L4 | Pulmonary TB/AFB+ | None | 2 | 888841 2047599 2445555 2947148 | Intergenic Intergenic Rv2183c Rv2619c | N/A N/A nsSNP nsSNP | 58 ND ND ND | 63 37 41 17 | 69 60 58 24 | N/A N/A CH CH | 8/2 | No relapse after 8 months anti-TB treatment |
| P15/L3 | Pulmonary TB/AFB+ | None | 2 | 1521699 3016079 | Rv1354c suhB Rv2701c | nsSNP sSNP | 93 30 | ND ND | ND ND | CH IMR | 7/6 | No relapse after 9 months anti-TB treatment |

P: Patient; L: Mtb lineage; AFB: Acid Fast Bacilli; RIF: Rifampicin; *in vitro* RIF exposure: representing a single experiment of 4 weeks of RIF exposure at 1 x minimum inhibitory concentration (MIC) in MGIT; N/A: Not Applicable

\* Variant frequency assessed by targeted NGS; ND: Not detected = Below Limit of Detection of WGS (coverage ranging from 46 to 185x) and targeted NGS (coverage ranging from 15 000 to 50 000x); Conserved hypotheticals: CH; Cell wall and cell processes: CWCP; Intermediary metabolism and respiration: IMR; Information pathway: IP; Lipid metabolism: LM; Regulatory proteins: RP; Virulence, detoxification, adaptation: VDA.

3280555 G) revealed by RIF exposure, (Table 1), we decided to focus on the last isolate of patient P1. To decipher the dynamics of the variant composition of the last isolate of patient P1, we first performed WGS on 20 single colonies isolated as described in the Material and Methods section. Apart from the predominant clone, 4 variants were identified and their presence in the last isolate of patient P1 was confirmed by targeted NGS, at frequencies between 0.5% and 9% (Fig 2A). Six loci differed among the variants, with a maximum pairwise distance of 4 SNPs. On the one hand, this isolate was exposed for 4 weeks to RIF, *in vitro*, at 1xMIC (Fig 2B) and on the other hand it was submitted to 7 days of macrophage infection before recovering intracellular bacteria (Fig 2C). Interestingly, NGS analysis found that the C 3280555 G variant was the most enriched during both RIF exposure (from 1.5% to 45%) and macrophage infection (from 1.5% to 74%), while the relative frequency of the other 3 variants varied within a range of 0.05 to 10% (Fig 2).

We further studied the dynamics of the diversity of the clinical isolate by focusing on its two main composing variants that we discriminated, according to WGS data, by the nucleotide 3280555 (i.e. G for the minor variant and C for the major variant). Two single-colony variants stemmed from the last isolate of patient P1 were isolated to be compared: the initially majority variant (IMV), carrying the C nucleotide at position 3280555, and the variant carrying the fixed C 3280555 G polymorphism. When appropriate, experiments were also performed on a 50:50 mixture of these two variants.

### The C 3280555 G harboring variant overexpresses mycocerosates and more particularly the tetramethyl-branched components of mycocerosates

The SNP C 3280555 G results in an amino-acid change, A721P, of mycocerosic acid synthase (Mas, Rv2940c). It is located in the Mas acyltransferase domain, which is involved in multi-methylated mycocerosates synthesis by iterative condensation of methyl-malonyl-CoA units [25,26]. Mycocerosates are components of the phthiocerol dimycocerosate (PDIM) and of phenolic glycolipids (PGL) of Mtb, both strongly involved in host-pathogen interaction [27–29]. Because the mycocerosate components are expected to be altered in the C 3280555 G variant, the lipid profile of this variant and of the IMV were explored, focusing on the production of the 4 main components of mycocerosates (Fig 3A): the two trimethyl-branched, C27 and

**Table 2. Mtb variants from control clinical isolates and emerging after 4 weeks of exposure to 1x MIC RIF *in vitro*.**

| Patient/ Mtb lineage | Clinical presentation/ smear | Comorbidity | Duration of positive culture, weeks | Time to first negative culture, weeks | Position | Gene | Mutation type | Variant frequencies (%) | | Functional categories | Severity score Bandim score/ MUST | Outcome (2 years follow-up) |
|---|---|---|---|---|---|---|---|---|---|---|---|---|
| | | | | | | | | Initial sample | *in vitro* RIF exposure | | | |
| C1/L1 | Pulmonary TB/AFB + | None | 2 | 8 | N/A | N/A | N/A | N/A | N/A | N/A | 6/5 | No relapse after 9 months anti-TB treatment |
| C2/L2 | Disseminated TB/AFB + | Hepatitis B | 3 | 4 | N/A | N/A | N/A | N/A | N/A | N/A | 7/6 | No relapse after 10 months anti-TB treatment |
| C3/L2 | Disseminate TB/AFB + | None | 3 | 8 | 46646 2592310 | Rv0042c Rv2319c | nsSNP sSNP | 41 72 | 25 84 | RP VDA | 7/6 | No relapse after 9 months anti-TB treatment |
| C4/L2 | Disseminated TB/ AFB + | Immuno-suppressive therapy | 4 | 6 | N/A | N/A | N/A | N/A | N/A | N/A | 2/4 | No relapse after 12 months anti-TB treatment |
| C5/L2 | Pulmonary TB/AFB + | None | 3 | 8 | N/A | N/A | N/A | N/A | N/A | N/A | 4/0 | No relapse after 6 months anti-TB treatment |
| C6/L3 | Pulmonary TB/AFB + | None | 2 | 3 | N/A | N/A | N/A | N/A | N/A | N/A | 2/3 | No relapse after 6 months anti-TB treatment |
| C7/L3 | Pulmonary TB/AFB + | None | 1 | 5 | 3342198 | mutT1 Rv2985 | nsSNP | **1.6**\* | 36 | IP | 3/2 | No relapse after 6 months anti-TB treatment |
| C8/L3 | Pulmonary TB/AFB + | None | 6 | 8 | N/A | N/A | N/A | N/A | N/A | N/A | 2/2 | No relapse after 6 months anti-TB treatment |
| C9/L4 | Disseminated TB/AFB + | Hepatitis B | 1 | 3 | 1603335 | fadD12 Rv1427c | sSNP | 81 | 97 | LM | 9/6 | No relapse after 6 months anti-TB treatment |
| C10/L4 | Pulmonary TB/AFB + | None | 1 | 2 | N/A | N/A | N/A | N/A | N/A | N/A | 4/2 | No relapse after 6 months anti-TB treatment |

(*Continued*)

**Table 2.** (Continued)

| Patient/ Mtb lineage | Clinical presentation/ smear | Comorbidity | Duration of positive culture, weeks | Time to first negative culture, weeks | Position | Gene | Mutation type | Variant frequencies (%) | | Functional categories | Severity score Bandim score/ MUST | Outcome (2 years follow-up) |
|---|---|---|---|---|---|---|---|---|---|---|---|---|
| | | | | | | | | Initial sample | *in vitro* RIF exposure | | | |
| C11/L4 | Pulmonary TB/AFB + | Diabetes | 5 | 8 | N/A | N/A | N/A | N/A | N/A | N/A | 3/3 | No relapse after 6 months anti-TB treatment |
| C12/L4 | Pulmonary TB/AFB + | Diabetes | 4 | 8 | N/A | N/A | N/A | N/A | N/A | N/A | 7/4 | No relapse after 6 months anti-TB treatment |
| C13/L4 | Pulmonary TB/AFB + | None | 3 | 4 | 1231745 | Rv1105 | sSNP | 67 | 70 | IMR | 11/4 | No relapse after 9 months anti-TB treatment |
| C14/L4 | Pulmonary TB/AFB + | Diabetes | 2 | 6 | N/A | N/A | N/A | N/A | N/A | N/A | 4/1 | No relapse after 9 months anti-TB treatment |
| C15/L4 | Disseminated TB/AFB + | None | 2 | 6 | 1548675 | Rv1375 | nsSNP | 46 | 37 | CH | 7/4 | No relapse after 9 months anti-TB treatment |
| C16/L4 | Pulmonary TB/AFB + | None | 3 | 4 | N/A | N/A | N/A | N/A | N/A | N/A | 7/4 | No relapse after 9 months anti-TB treatment |
| C17/L4 | Pulmonary TB/AFB + | None | 4 | 8 | N/A | N/A | N/A | N/A | N/A | N/A | 8/6 | No relapse after 9 months anti-TB treatment |
| C18/L4 | Pulmonary TB/AFB + | None | 0 | 1 | 3685762 | atsB Rv3299c | nsSNP | ND | 40 | IMR | 2/0 | No relapse after 6 months anti-TB treatment |
| C19/L4 | Pulmonary TB/AFB + | None | 2 | 4 | N/A | N/A | N/A | N/A | N/A | N/A | 6/4 | No relapse after 6 months anti-TB treatment |
| C20/L4 | Pulmonary TB/AFB + | Immuno-suppressive therapy | 1 | 3 | N/A | N/A | N/A | N/A | N/A | N/A | 3/2 | No relapse after 6 months anti-TB treatment |

C: Patient of control cohort; L: Mtb lineage; AFB: Acid Fast Bacilli; RIF: Rifampicin; *in vitro* RIF exposure: representing a single experiment of 4 weeks of RIF exposure at 1 x minimum inhibitory concentration (MIC) in MGIT; N/A: Not Applicable

* Variant frequency assessed by targeted NGS; ND: Not detected = Below the limit of detection of WGS (coverage ranging from 45 to 111x) and targeted NGS (coverage ranging from 18 000 to 57 000x); Conserved hypotheticals: CH; Intermediary metabolism and respiration: IMR; Information pathway: IP; Lipid metabolism: LM; Regulatory proteins: RP; Virulence, detoxification, adaptation: VDA.

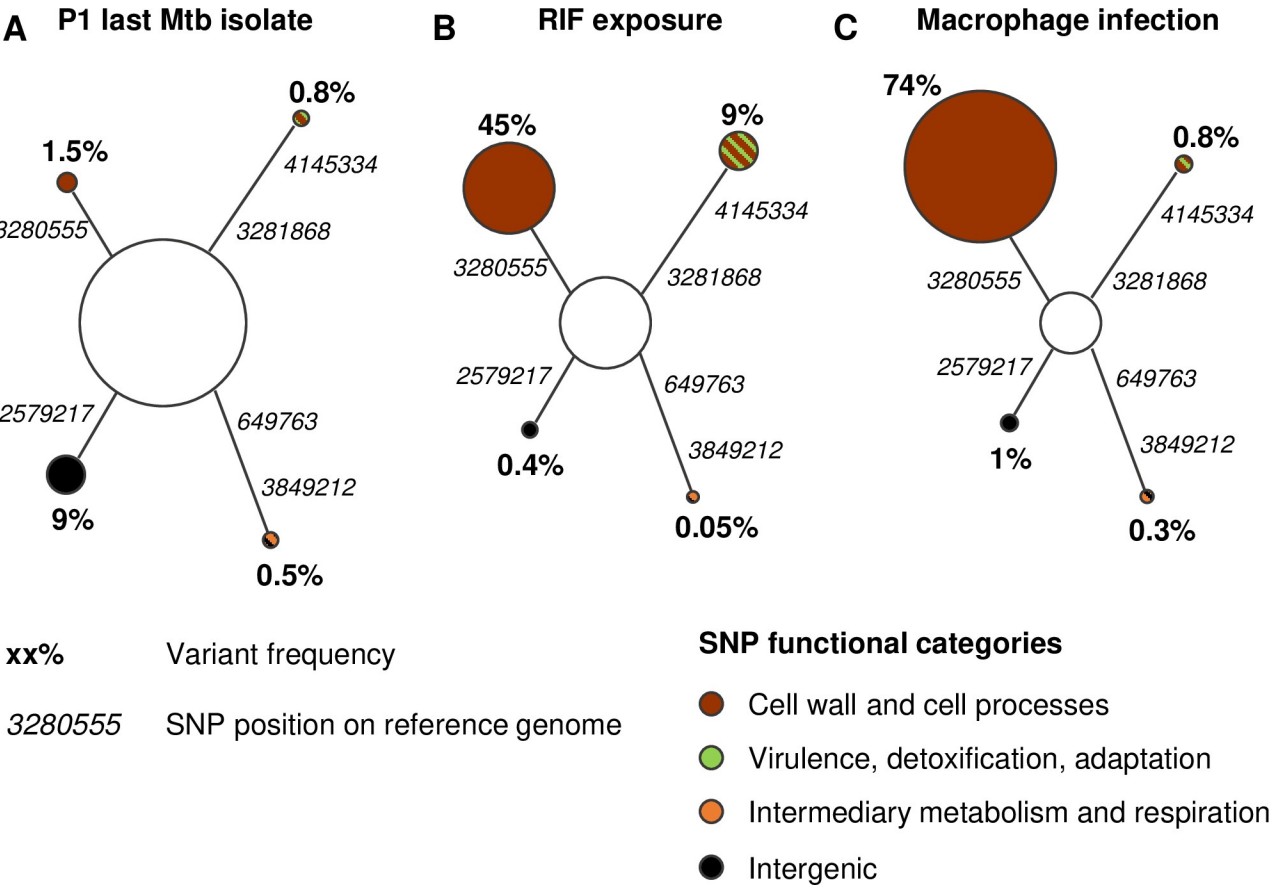

**Fig 2. A minor variant of Mtb clinical isolate is selected in response to both rifampicin (RIF) exposure and macrophage infection.** Minimum spanning trees (MST) of (A) the last Mtb isolate of P1 patient, (B) after *in vitro* RIF exposure or (C) after 7 days of macrophage infection, assessed by targeted NGS with coverage ranging from 10 000 to 36 000x. (B) After 4 weeks of *in vitro* RIF exposure at 1x minimum inhibitory concentration (MIC) in MGIT, bacteria were plated and then sequenced. (C) Macrophages were infected by the last isolate of P1 patient at a multiplicity of infection (MOI) of 10:1 (bacteria:cells). At 7 days post-infection, macrophages were lysed to recover, plate and then sequence intracellular bacteria. All experiments were performed at least twice, and MST are pooled results of these independent experiments.

C29, and the two tetramethyl-branched, C30 and C32 [30,31]. When compared to the IMV, the C 3280555 G variant profile showed significantly lower trimethyl-branched C27 component (11.4 ± 2.2%) than tetramethyl-branched C30 component (80.1 ± 2.4%; Fig 3B), consistent with a higher affinity of Mas enzyme for methyl-malonyl-CoA that promotes tretramethyl mycocerosate synthesis in this variant. Moreover, the relative proportion of mycocerosates produced by the C 3280555 G variant was approximately twice as high as that produced by the IMV (Fig 3C). Given this lipid profile, the variant carrying the C 3280555 G polymorphism is hereafter referred to as the 4MBE (tetra-methyl branched enriched) variant.

## The 4MBE variant is tolerant to RIF alone and in combination with isoniazid

As the 4MBE variant was revealed upon RIF exposure its susceptibility to this drug was explored. Although there was no difference regarding the MIC values (0.12 mg/L for both 4MBE and IMV), the tolerance of the 4MBE variant to sub-MIC doses of RIF was investigated. The kinetics of the bacterial growth, explored by the time to positivity (TTP) approach, revealed similar growth profiles in the absence of RIF (from 6.7 ± 0.3 days to 7.2 ± 0.3 days),

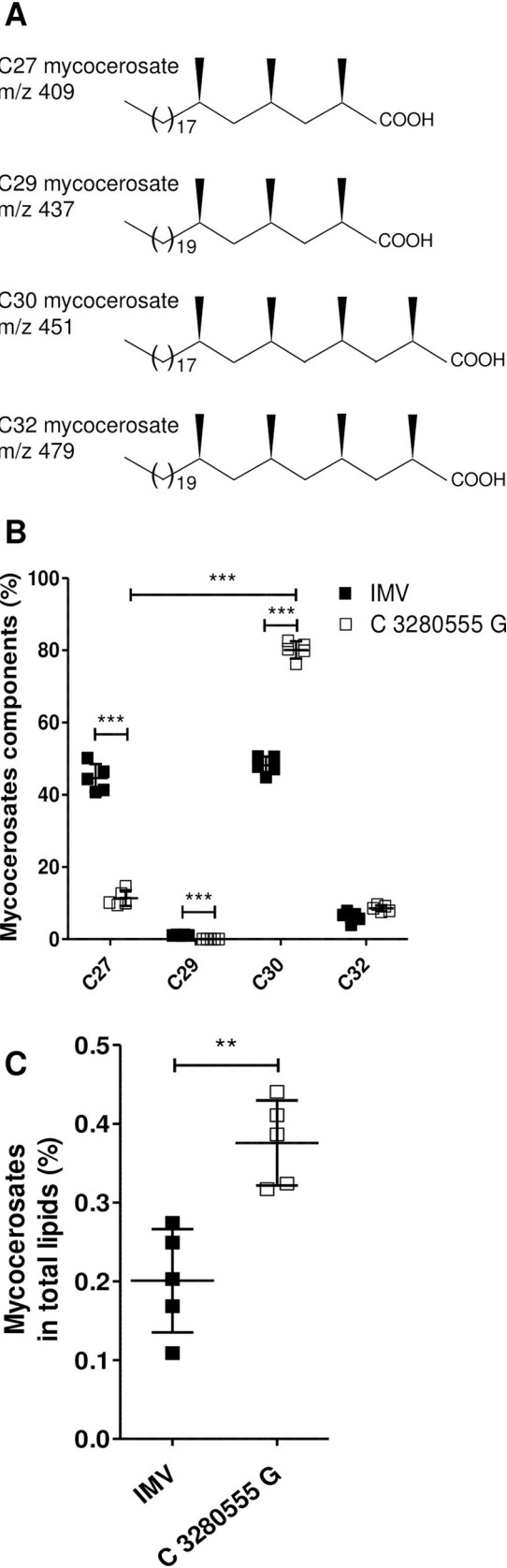

**Fig 3. The 4MBE variant overexpresses mycocerosates and more particularly the tetramethyl-branched components of mycocerosates.** (A) Molecular structures of C27, C29, C30 and C32 mycocerosate components from Mtb. (B-C) The lipid profile of the initially majority variant (IMV, black) and the C 3280555 G variant (white) was assessed. (B) Proportion of mycocerosate components regarding total mycocerosates were explored by gas chromatography-mass spectrometry (GC-MS) and obtained from ion chromatograms showing the fragment ions at m/z 409, 437, 451 and 479 representing mycocerosate components C27, C29, C30 and C32, respectively. (C) Proportion of mycocerosates in total lipids were obtained from ion chromatograms showing all fatty acids detected by GC-MS. Values for each condition are the mean ± standard deviation (SD) of five independent experiments. Means were compared using One-Way ANOVA followed by Bonferroni correction (B) or using unpaired t-test (C). $^{*}p<0.05$, $^{**}p<0.01$, $^{***}p<0.001$.

showing the same fitness of these variants in the absence of antibiotic pressure. Conversely, a significantly shorter TTP for the 4MBE variant (7.1 ± 0.4 days) compared to the IMV (15.6 ± 1.3 days) was observed upon antibiotic treatment with 1xMIC of RIF (Fig 4A), suggesting a better ability of the 4MBE variant to persist upon RIF exposure.

To investigate more deeply the antibacterial effect of RIF against the IMV and the 4MBE variant, concentration-effect experiments were performed, and a pharmacological efficacy prediction model was then applied to the results obtained. While there were no significant differences for the maximal effect of RIF, there was a decrease in RIF susceptibility for the 4MBE variant at sub-MIC doses (Fig 4B). The 4MBE variant exhibited higher $EC_{50}$ value (5.9, 95% CI [3.4; 10.4]) than the IMV (1.0, [0.6; 1.6]), suggesting a lower potency of this antibiotic on 4MBE variant (S1 Table). Overall, these results indicate that the 4MBE variant is more tolerant to sub-MIC doses of RIF compared to the IMV, despite the absence of canonical RIF resistance.

To further explore RIF tolerance, we performed time-kill experiments upon RIF exposure at 2, 4, 8, and 16xMIC RIF to compare the minimum duration for killing 99% of the mycobacterial population ($MDK_{99}$) for the IMV and the 4MBE variant. A higher $MDK_{99}$ was observed for the 4MBE variant compared to IMV for each dose of RIF tested (Figs 4C and S2). At the lowest dose of RIF tested (2xMIC), the $MDK_{99}$ observed for the 4MBE variant was greater than 10 days while the mean $MDK_{99}$ for IMV was 7.6 ± 0.5 days. For the highest dose of RIF tested (16xMIC) the mean $MDK_{99}$ for the 4MBE variant remained significantly greater than the $MDK_{99}$ for the IMV (5.1 ± 0.4 days versus 3.1 ± 0.2 days, $p = 0.0018$). Taken together, these results confirm that the 4MBE variant is more tolerant to RIF than the IMV.

As RIF is part of a complex regimen during TB treatment, the antibacterial effect of the combination of both first-line anti-TB drugs (isoniazid [INH] and RIF) was investigated on the IMV and the 4MBE variant (Fig 4D). The Minto model described the combined antibacterial effect very well, $R^2$ values ≥ 0.90 and low bias and imprecision values (S2 Table). For RIF an almost 6-fold higher $EC_{50}$ value was observed for the 4MBE variant compared to the IMV. For INH, the 4MBE variant exhibited a slightly higher $EC_{50}$ value (0.91 [0.80; 1.03]) than the IMV (0.53 [0.48–0.58]), and exhibited also a lower Hill coefficient value (2.07 [1.74; 2.41]) than the IMV (3.54 [2.64, 4.43]), suggesting also a lower potency of INH on the 4MBE variant. More importantly, the interaction parameter $\beta_{U50}$ was significantly greater than 0 for the IMV (1.43 [0.94; 1.91]), which means that the combined effect of INH and RIF was synergistic. Conversely, it was not significantly different from 0 for the 4MBE variant (0.68 [-0.38; 1.74]), indicating that the combined effect was only additive. Taken together, these results indicate that the 4MBE variant have increased tolerance to RIF and INH, alone but also in combination, possibly favoring treatment failure.

To explore a possible selective advantage of the 4MBE variant upon treatment, the dynamics of variant assemblies upon RIF exposure was monitored. Mycobacteria in a IMV:4MBE 50:50 mixture were recovered after 4 weeks of growth in MGIT, with or without 1xMIC of

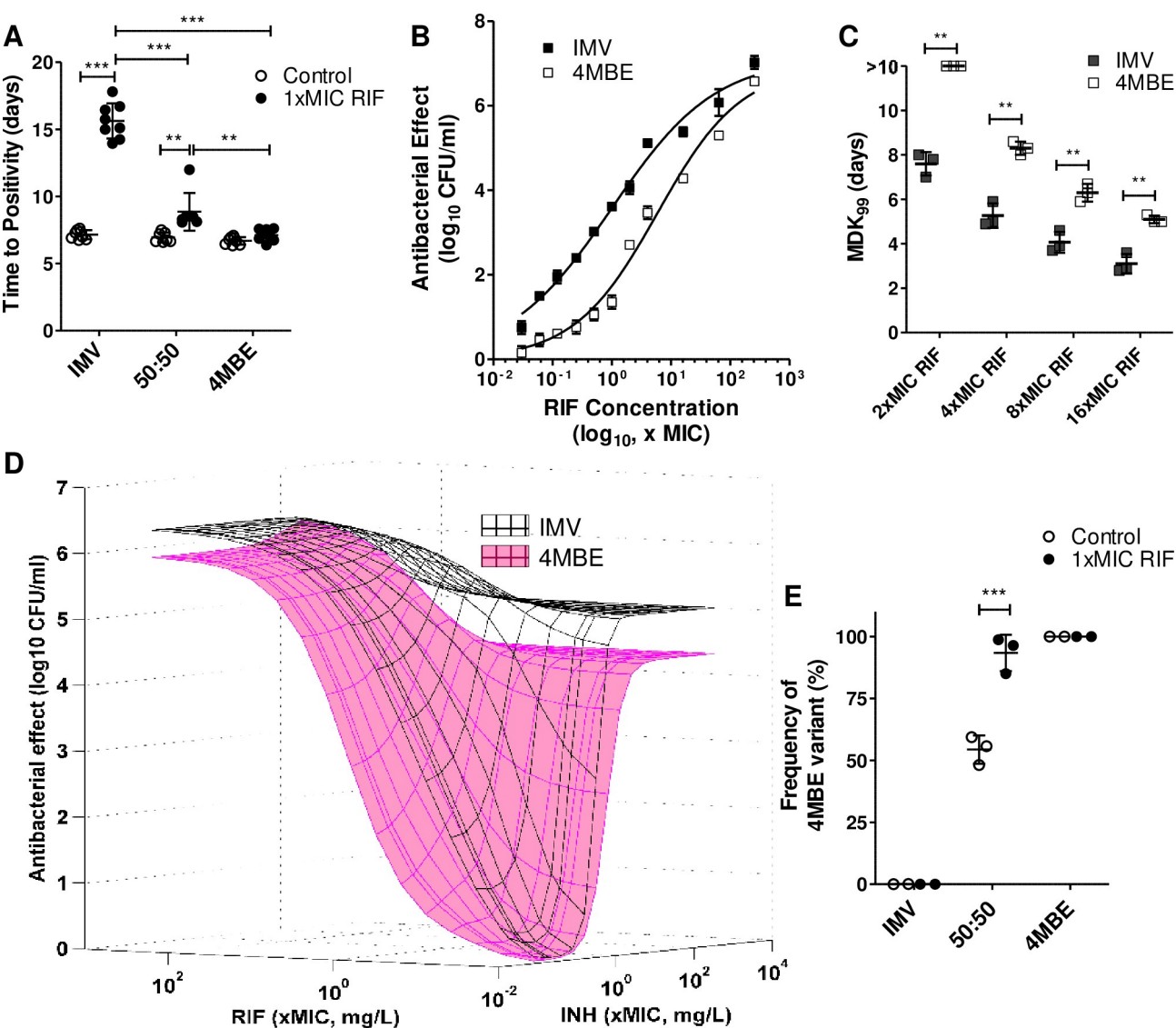

**Fig 4. The 4MBE variant is tolerant to rifampicin (RIF) alone and in combination with isoniazid (INH).** (A) Mycobacterial growth of the IMV, 4MBE variant, and a 50:50 mixture in absence (white; Ctrl, inoculated with $10^3$ CFU/mL) or presence of RIF (black; 1x minimum inhibitory concentration (MIC) RIF, inoculated with $10^5$ CFU/mL) is expressed in time to positivity (TTP) in the BACTEC system. Values for each condition are the mean ± standard deviation (SD) of seven or eight independent experiments. Means were compared using one-way ANOVA followed by Bonferroni correction. (B) Time-kill data were determined after 7 days of RIF exposure, at concentrations ranging from 1/32 to 256xMIC, for the IMV (black) and the 4MBE variant (white). Fit of the Hill pharmacodynamic model to rifampicin (RIF) time-kill data, the solid lines are the best fit lines. The y-axis represents the difference in $\log_{10}$ CFU/mL relative to control (C = 0) from a duplicate experiment. (**C**) Inocula of IMV and 4MBE variant at $2.10^7$ CFU/mL were exposed to 2, 4, 8, and 16xMIC of RIF. Mycobacterial survival was determined at 3, 5, 7, and 10 days post-exposure by CFU counting. Variation in RIF tolerance was determined by the minimum duration for killing 99% of the mycobacterial population (MDK$_{99}$) time. Values for each condition are the mean ± SD of three independent experiments. Means were compared using Student's t-test. (D) Superposition of response surface models of the combined antibacterial effect of INH and RIF, from 1/32 to 256xMIC, against the IMV (black) and the 4MBE variant (pink). A total of 142 measures of the effect of RIF and INH alone or in combination were performed, and results were analyzed using the Minto response surface model. The surfaces represent the model-based predicted effect, representing the antibacterial effect measured after 7 days of exposure. (E) After 4 weeks of incubation without (white; Ctrl) or with RIF (black; 1xMIC RIF), the change in the 4MBE variant frequency on plated bacteria was assessed by droplet digital PCR (ddPCR) as described in the Material and Methods section. Values for each condition are the mean ± standard deviation (SD) of two or three independent experiments. Means were compared using the Student's t-test. $^*p<0.05$, $^{**} p <0.01$, $^{***} p <0.001$.

RIF, to extract DNA after plating and assess the frequency of the 4MBE variant by droplet digital PCR (ddPCR) thanks to the C 3280555 G polymorphism (Fig 4E). Without antibiotic pressure, the distribution of the variants in the 50:50 mixture was substantially stable (54.4 ± 5.7%) after 4 weeks of culture. Conversely, after RIF exposure, the 4MBE variant became predominant (93.4 ± 7.3%), consistent with the previous NGS experiments (Fig 2B).

### Improved Mtb intra-macrophagic survival of the 4MBE variant

As the 4MBE variant was also revealed upon macrophage infection (Fig 2C), the abilities of intra-macrophage survival of the IMV, the 4MBE variant and their 50:50 mixture were explored at 6 h and 96 h post-infection (hpi).

At 6 hpi, the 4MBE variant uptake by macrophages was significantly better ($28.1 \times 10^4 \pm 9.0 \times 10^4$ CFU/mL) than that of the IMV ($4.7 \times 10^4 \pm 2.4 \times 10^4$ CFU/mL, Fig 5A). Moreover, the intracellular growth ratio between 96 hpi and 6 hpi was significantly better for the 4MBE variant (5.0 ± 0.9 -fold) than for the IMV (1.4 ± 0.4 -fold), indicating a better intra-macrophagic survival and/or multiplication (Fig 5B).

To further monitor the relative proportion between the variants upon host-pathogen interaction, the DNA of plated bacteria at 6 hpi and 96 hpi were recovered to assess the variant frequency by ddPCR (Fig 5C). An enrichment of the 4MBE variant, up to 75.0 ±10.7% at 96 hpi, was observed, consistent with the previous NGS experiments (Fig 2C).

### Improved intra-macrophagic survival of the 4MBE variant is due to phagolysosome avoidance and inhibition of autolysosome formation

Mtb intramacrophagic behavior of the IMV and the 4MBE variant was explored by confocal microscopy analysis at 24 hpi (Fig 6). The improved Mtb intra-macrophagic survival upon infection with the 4MBE variant compared to the IMV was confirmed by a significantly greater number of both infected cells and bacterial burden per cell (S3 Fig).

To explore the mechanisms of intra-macrophage survival of the 4MBE variant, phagolysosome activation and avoidance were first explored (Fig 6A–6C). Although no difference was observed in the number of cells with acidified compartments (Fig 6A), there were significantly

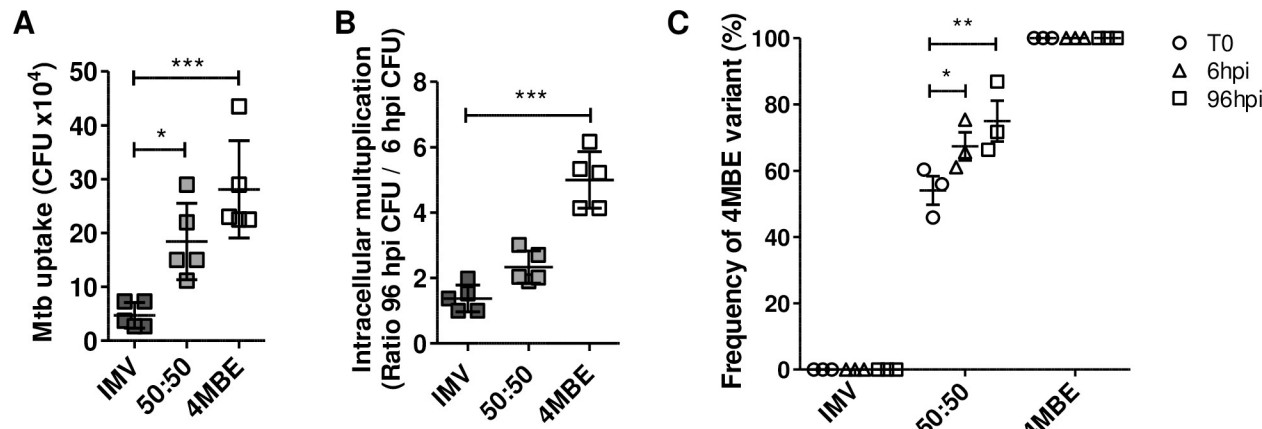

**Fig 5. Improved Mtb intra-macrophagic survival of the 4MBE variant.** Macrophages were infected by the IMV (black), the 4MBE variant (white) or a half-half mixture (50:50, grey) at a multiplicity of infection (MOI) of 10:1. (A-B) At 6 and 96 hours post-infection (hpi), macrophages were lysed for intracellular CFU counting to study (A) Mtb uptake by macrophages thanks to intracellular CFU counting at 6 hpi and (B) intracellular multiplication thanks to intracellular CFU ratio between 6 hpi and 96 hpi. (C) The change in the 4MBE variant frequency on plated bacteria was assessed by ddPCR as described in the Material and Methods section. Values for each condition are the mean ± SD of three to five independent experiments. Means were compared using One-Way ANOVA followed by Bonferroni correction. $^*p<0.05$, $^{**}p<0.01$, $^{***}p<0.001$.

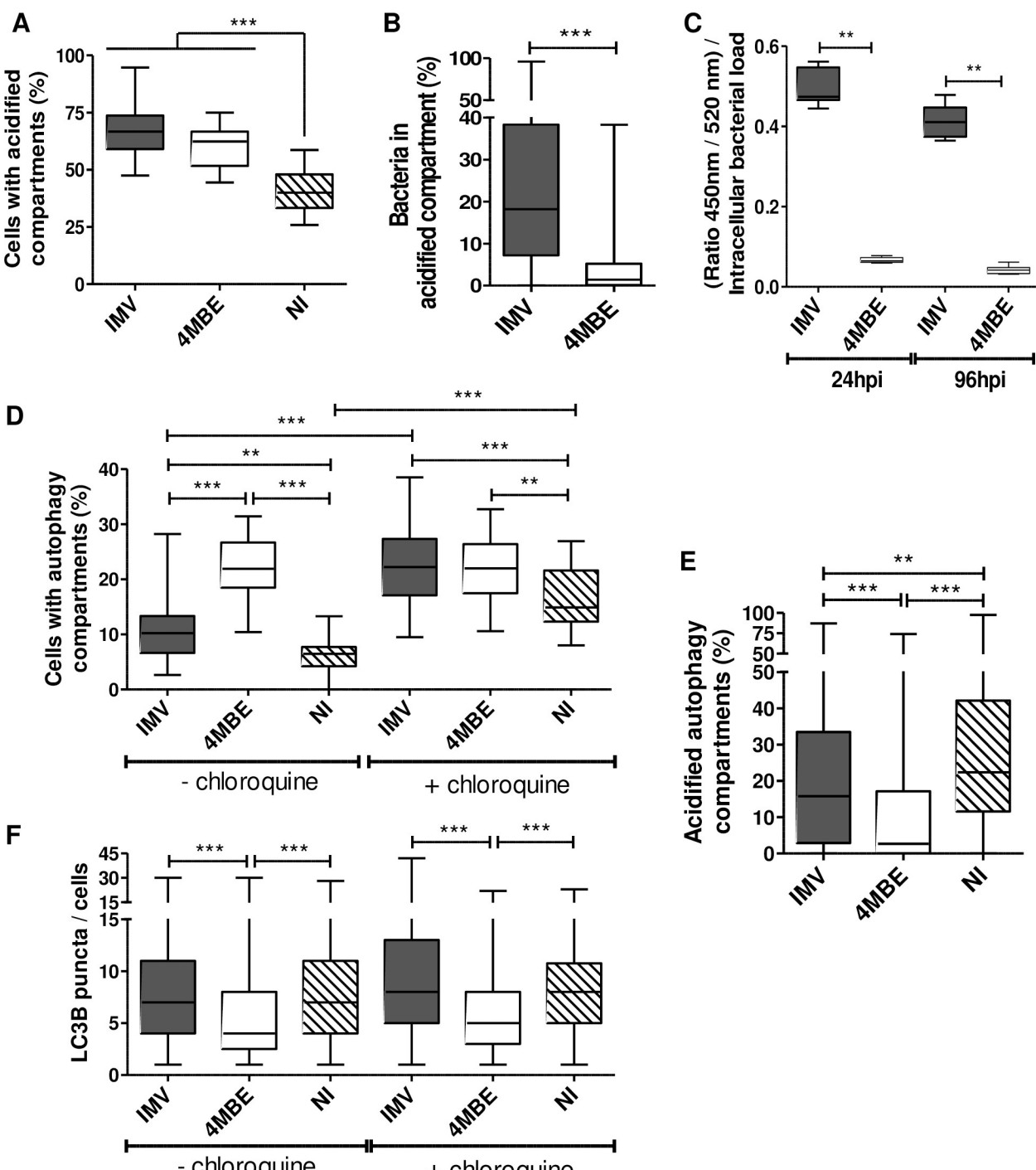

**Fig 6. Improved intra-macrophagic survival of the 4MBE variant is due to phagolysosome avoidance and inhibition of autolysosome formation.** Macrophages were infected by the IMV (black) or the 4MBE variant (white) at a MOI of 10:1 or not infected (NI; hatched) as a control. (A-B and D-F) At 24 hpi, cells were stained with LysoTracker Red DND-99, anti-Mtb coupled FITC, and anti-LC3B coupled Dy-Light 650 antibodies. Cells with acidified (A) or autophagy compartments (D) were determined across at least 10 confocal images, containing between 30 and 80 cells per image, per replicate. Proportion of bacteria in acidified compartments (B), proportion of acidified autophagy compartments (E), and LC3B puncta per cells (F) were determined across at least 40 cells per replicate. (D and F) Three hours before staining, cells were incubated with chloroquine, then macrophages were stained and analyzed as previously described. (A-B and D-F) Values for each condition are the median values [interquartile range, IQR] of at least three independent experiments. Statistical significance was determined using Kruskal-Wallis analysis, using Dunn's Multiple Comparison Test or Mann Whitney test, where appropriate. (C) Infected macrophages were loaded with CCF4-AM at 24 and 96 hpi to analyze 450nm/520nm ratio normalized to bacterial load. Values for each condition are the mean ± standard deviation (SD) of at least three independent experiments. Means were compared using Mann Whitney test. *$p < 0.05$, ** $p < 0.01$, *** $p < 0.001$.

more bacteria in acidified compartments upon infection by the IMV (18.2% [7.1–38.3]) compared to the 4MBE variant (1.4% [0.03–5.2], Fig 6B). To determine whether this decreased colocalization of Mtb with acidified compartments was due to Mtb translocation from the phagolysosome to the cytosol or to an inhibition of phagolysosomal fusion, infected macrophages were loaded with CCF4-AM, a fluorescence resonance energy transfer (FRET)-based assay. After normalization on the intracellular bacterial load, the blue/green signal was almost 10 times lower upon infection by the 4MBE variant than the IMV at 96 hpi (Fig 6C). This supports the notion that the 4MBE variant avoids phagolysosomal fusion rather than translocating to the cytosol.

The autophagy activation and outcome were then explored. Although the proportion of cells with autophagy compartments was approximately two-fold upon infection by the 4MBE variant compared to the IMV, chloroquine treatment abolished the differences between the infecting variants. Moreover, the presence or absence of chloroquine pretreatment did not modify the proportion of cells with autophagy compartments upon infection by the 4MBE variant, suggesting an inhibition of the autolysosome formation by this variant (Fig 6D). This was confirmed by the significant decrease in the proportion of acidified autophagy compartments upon infection by the 4MBE variant compared to the IMV (Fig 6E). Furthermore, there was a decrease in the number of LC3B puncta per cells upon infection by the 4MBE variant, of approximately 1.5-fold, compared to the IMV (Fig 6F), indicating a weaker autophagy activation upon infection by the 4MBE variant.

Taken together, these results support that the better intracellular survival of the 4MBE variant during macrophage infection is driven by the increase of phagolysosome avoidance and the decrease of autophagy activation and outcome.

## Macrophage apoptosis and inflammatory response are exacerbated upon infection by the 4MBE variant

Macrophage transcriptome analysis found that 342 genes were differentially expressed between macrophages infected by the IMV and those infected by the 4MBE variant. A total of 228 differentially expressed genes were selected to build a heatmap, showing clustering according to biological replicates (Fig 7). The differentially expressed genes were classified into nine functional categories involved in Mtb macrophage response. Consistent with the macrophage infection experiments, actin cytoskeleton, autophagy, and phagolysosome pathways were found to be dysregulated upon infection by the 4MBE variant compared to the IMV. Moreover, there was also a dysregulation of inflammatory response, apoptosis and cell cycle, mitochondrial metabolism, lipid metabolism, and T cell signaling pathways.

In order to explore cell death mechanisms, lytic cell death (Fig 8A) and apoptosis (Fig 8B) of macrophages were assessed upon infection by the IMV and the 4MBE variant at 24 and 96 hpi, as apoptosis is generally regarded as a protective response, whereas lytic cell death is thought to favor inflammation and disease progression [32–34]. At 96 hpi, the lytic cell death upon 4MBE variant infection (64.9 ± 3.5%) was significantly greater than that obtained upon IMV infection (Fig 8A). Furthermore, apoptosis at 96hpi was also significantly higher upon infection by the 4MBE variant (15.1% [11.1–19.2]) compared to the IMV (7.9% [4.7–12.0], Fig 8B).

Cytokine (TNF-α, IL-6, IL-1β, IFN-γ), chemokine (RANTES (CCL5), and interferon gamma-induced protein 10 (IP-10) production were then measured at 6 hpi (S4 Fig), 24 hpi (Fig 9), and 96 hpi (S5 Fig) to explore macrophage inflammatory response. At 24hpi, there was a significant increase in the inflammatory response of macrophages upon infection by the 4MBE variant compared to the IMV (Fig 8). Upon infection by the 4MBE variant there was a

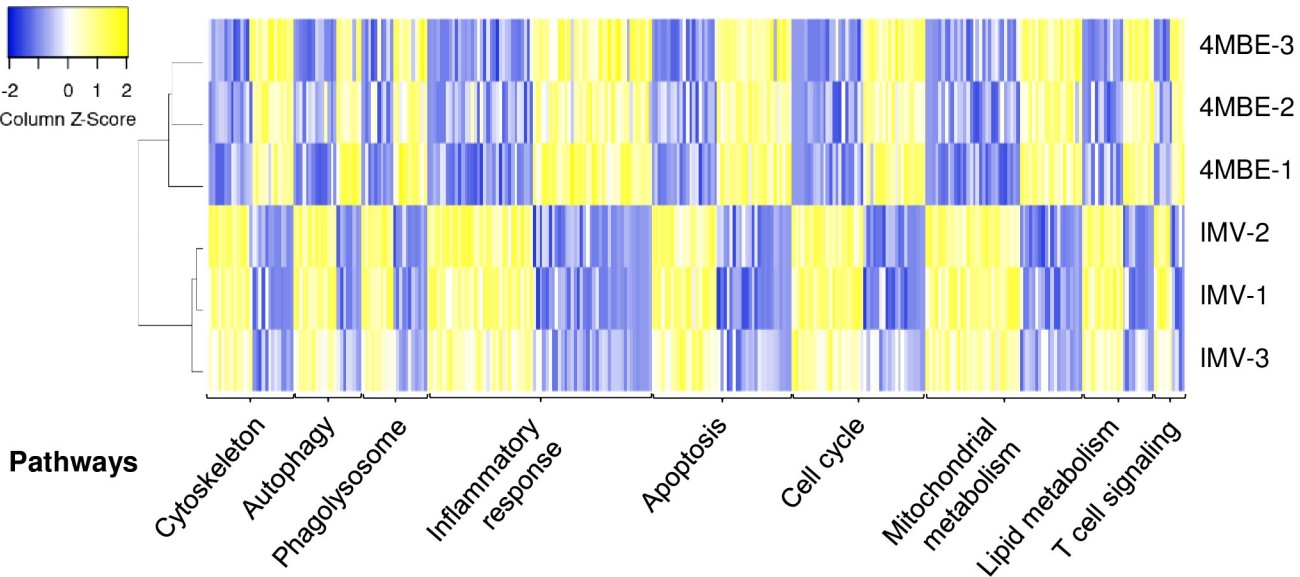

**Fig 7. Dysregulation of macrophage gene expression upon infection by the 4MBE variant.** Macrophages were infected at a MOI of 10:1 by the IMV or the 4MBE variant. At 24 hpi, total RNA were extracted to study the changes in global transcriptome of host cells. Heatmap of gene expression (log2 fold change) for 228 differentially expressed genes between IMV or 4MBE variant- infected macrophages at 24 hpi. Rows are independent experiments and columns are genes. Differentially expressed genes were classified in nine functional categories. Clustering was based on Pearson's correlation. The two experimental conditions clustered with their biological replicates.

significant increase in TNF-α production by macrophages from 6 to 96 hpi (Figs 9A, S4A and S5A) and overall a sustained inflammatory response at 96 hpi (S5 Fig). Finally, chemokine signals were significantly increased upon infection by the 4MBE variant (Fig 9E and 9F). Overall, the macrophage pro-inflammatory response was stronger (in terms of intensity and duration) upon infection by the 4MBE variant than by the IMV.

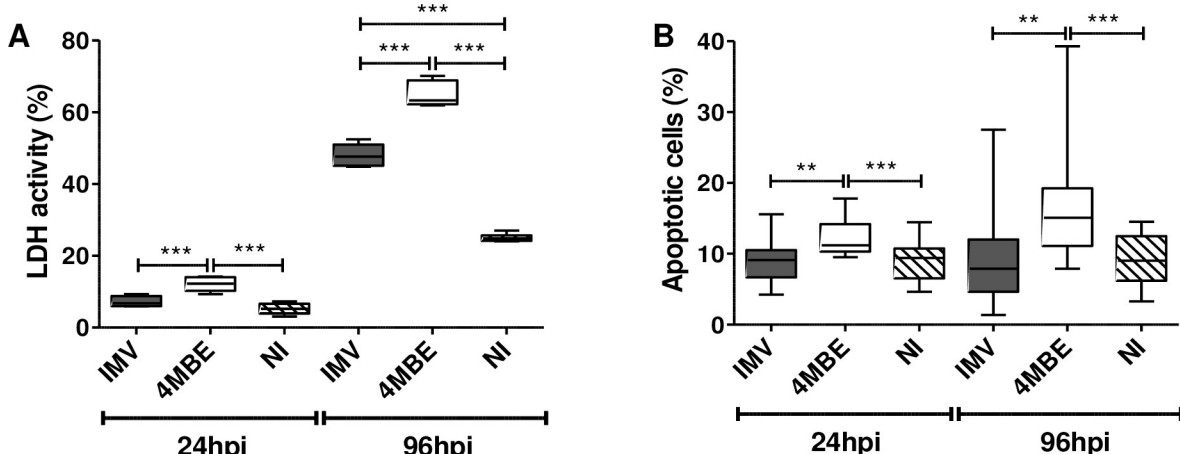

**Fig 8. Exacerbated macrophage cell death upon infection by the 4MBE variant.** Macrophages were infected by the IMV (black) or the 4MBE variant (white) at a MOI of 10:1 or not infected (NI; hatched) as a control. (A) Lytic cell death, at 24 hpi and 96 hpi, was evaluated by LDH release in cell culture supernatant from at least 3 independent experiments analyzed in duplicate. Means were compared using One-Way ANOVA followed by Bonferroni correction. (B) Apoptosis was measured by Annexin V staining and analysis performed by microscopy, at 24 hpi and 96 hpi. Apoptosis was determined across at least 5 images, containing between 20 and 50 cells per image, per replicate, in at least three independent experiments. Statistical significance was determined using Kruskal-Wallis analysis, using Dunn's Multiple Comparison Test. $^*p<0.05$, $^{**}p<0.01$, $^{***}p<0.001$.

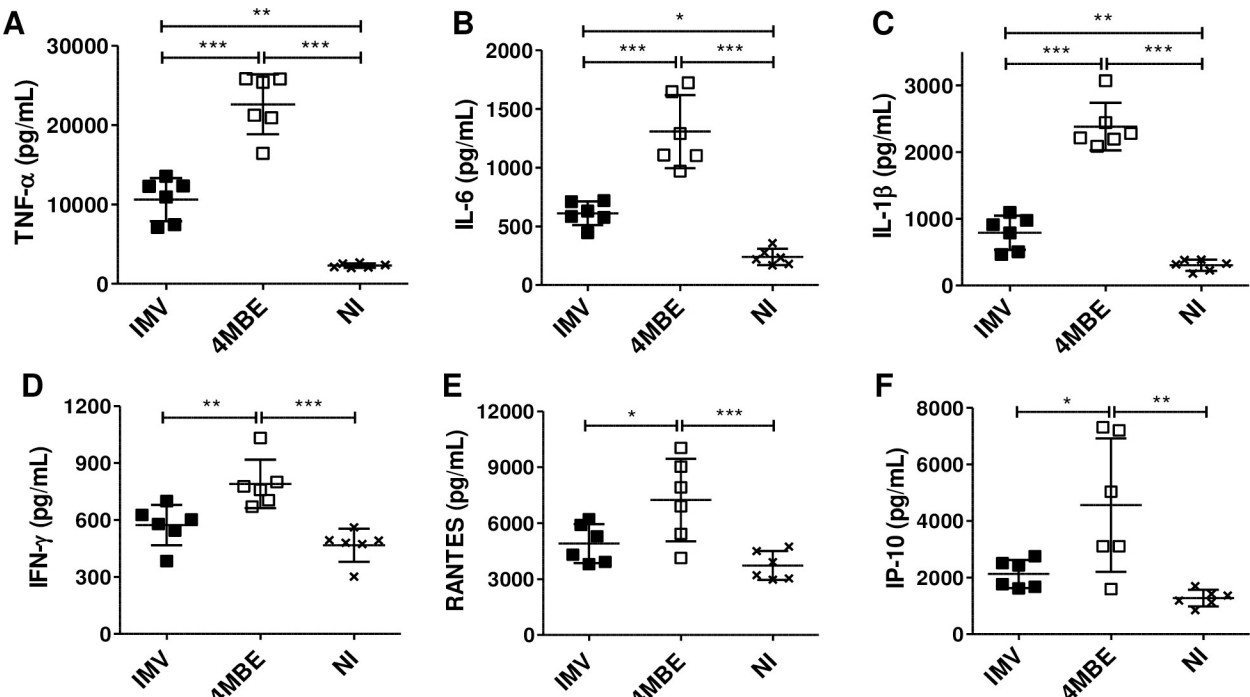

**Fig 9. Macrophage inflammatory response is exacerbated upon infection by the 4MBE variant.** Macrophages were infected by the IMV (black) or the 4MBE variant (white) at a MOI of 10:1 or not infected (NI, cross). (A) TNF-α, (B) IL-6, (C) IL-1β, (D) IFN-γ, (E) RANTES (CCL5) and (F) Interferon gamma-induced protein 10 (IP-10) release in cell culture supernatant was evaluated by Luminex Multiplex assay at 24 hpi. Values for each condition are the mean ± SD of three independent experiments, analyzed in duplicate. Means were compared using Repeated Measures ANOVA followed by Bonferroni correction. *$p < 0.05$, ** $p < 0.01$, *** $p < 0.001$.

It should be noted that both increased cell death and higher inflammatory response revealed by these bulk infection data could be, at least partially, related to the higher bacterial load upon infection by the 4MBE variant.

## Discussion

The objective of the present study was to address the issue of within-host Mtb micro-diversity and its possible link with anti-TB treatment response. We therefore followed within-host evolution of Mtb clinical isolates in a cohort of patients with delayed culture conversion, along with a cohort of fast culture conversion patients. We focused on minor variants detected by unfixed SNP, with a particular interest in drug tolerant sub-populations revealed after RIF exposure. The results support the absence of 2-month culture conversion as a predictive factor of poor outcome [22] as, among the 15 patients with delayed culture conversion, 3 died and the median duration of treatment in order to prevent failure was of 9 months, IQR [8.25–9.75]. By contrast among the 20 fast-converting patients, there was no casualty, and the median duration of treatment was significantly decreased at a median of 6 months, IQR [6–9], $p = 0.05$. Interestingly, thanks to WGS, we were able to detect unfixed SNPs within the initial and/or last Mtb isolates for 9/15 delayed-conversion patients. When focusing on unfixed nsSNP, micro-diversity of Mtb clinical isolates was detected in 8/15 patients (equally distributed between early and last isolates). Although 9 additional unfixed nsSNP from 6/15 isolates were revealed by RIF *in vitro* exposure, none was linked to drug resistance, consistent with previous reports on within-host Mtb evolution [12,14,18,35]. Of note, in 2 isolates unfixed SNPs became fixed over the course of TB disease, while in 2 other isolates unfixed SNPs became

undetectable by WGS in the last Mtb isolate, consistent with previous observations that numerous variants emerge and then disappear during the course of infection and treatment within individual patients [18,35,36]. By contrast, in the cohort of fast-converting patients a lower proportion of minor genetic variation was observed in the initial isolate and only 2 additional nsSNP from 2/20 isolates were revealed by RIF exposure. Interestingly, when focusing on persistent infections, WGS of isolated colonies from a last Mtb isolate allowed detection of additional unfixed SNPs, previously undetected by WGS performed on the whole bacterial population, suggesting that more knowledge concerning Mtb diversity will gained in the future through the use of deeper sequencing.

Furthermore, we highlighted the plasticity of Mtb genetic micro-diversity of a clinical persistent isolate, which presented a changing pattern of variant assembly depending on the experimental setting (intra-macrophage infection, antibiotic stress). Interestingly, a variant (4MBE variant) which was present at low frequency in the initial Mtb clinical isolate, became predominant under both stress conditions tested. Of note, although the search of public databases (ReseqTB, phyTB, GMTVD) did not identify this fixed polymorphism in other Mtb genomes, as the quality of available archives is often insufficient to detect minor variants, we were able to detect it, at a frequency (~1%) undetectable by WGS, in another Mtb clinical isolate by performing high-sensitivity targeted ddPCR assay. Among the 140 clinical isolates tested by ddPCR screening, this particular isolate was multi-drug resistant (MDR) belonging to the Euro-American lineage.

Using a set of complementary approaches, we were able to show that this type of variant displayed tolerance to anti-TB treatment and improved intra-macrophagic persistence. We found that the 4MBE variant features a particular mycocerosates profile. Mycocerosates are components of PDIM and PGL, both playing important roles in Mtb host-pathogen interactions at many pathophysiological steps [27–29,37,38]. We also showed that the 4MBE variant has increased ability to infect macrophages, and to induce both macrophage inflammatory response and cell death (mainly apoptosis), possibly deleterious *in vivo* [32,34]. This pro-inflammatory profile could mainly favor infection initiation, and be less prone to develop in long-term infection, explaining the low frequency of the variant in the clinical isolates. Current diagnostic tools are based on the detection of fixed or unfixed variants in loci involved in drug resistance, to adapt treatment as early as possible [39,40]. Moreover, features associated with Mtb tolerance have been suggested to be potential pre-resistant markers [41,42], as drug-tolerance and persistent infection may foster drug-resistance emergence [5]. Notwithstanding that C 3280555 G would be a hitchhiking SNP, this study shows that a minor variant may be associated with drug-tolerance and promote treatment failure as predicted by pharmacological modelling. The mechanisms behind this tolerant phenotype should be deciphered to better understand the impact of such variants. A thorough study of such variants not necessarily linked to canonical drug-resistance, and their ability to promote anti-TB drugs tolerance may be crucial to prevent the subsequent emergence of resistance. Detecting unfixed antibiotic tolerant variants in Mtb clinical isolates, could thus provide new biomarkers to guide TB management in patients with treatment failure risk.

Interestingly, *in vitro* RIF exposure revealed the presence of a low-frequency variant within the Mtb bacterial population, otherwise undetected by the WGS bulk analysis. From an evolutionary perspective, given the lack of on-going horizontal gene exchange in Mtb, genetic micro-diversity, including minor variants with particular phenotypic profiles, could be envisioned as a pathogen feature required to effectively respond to changing environments, to persist, and to spread [18,35,36,43,44]. This emphasizes the importance of Mtb micro-diversity, which could become a useful biomarker to predict, in association with previously proven factors, which patients are at risk of poor response to anti-TB treatment. These patients may

require personalized dosing and therapeutic management in order to improve the outcome and to effectively prevent treatment failure and resistance emergence.

## Material and methods

### Ethics statement

Patient and microbiological data were recorded in accordance with decision 20–216 of the ethics committee of the Lyon University Hospital, France, and the French Bioethics laws (Reference methodology MR-004 that covers the processing of personal data for purposes of study, evaluation or research that does not involve the individual). Relevant approval regarding access to patient-identifiable information are granted by the French data protection agency (*Commission Nationale de l'Informatique et des Libertés*, CNIL). All study participants gave their informed consent (written or not) or non-opposition to participation, in line with French legal guidelines.

### Study design

Patients were retrospectively enrolled among the 435 patients managed for drug-susceptible TB at Lyon university hospital between 2014 and 2017. Fifteen patients with the following inclusion criteria were selected: positive *Mtb* culture at least 2 months after the beginning of anti-TB treatment without culture conversion between time-points and complete follow-up 2 years after treatment completion. For the control cohort, 20 patients with the following inclusion criteria were selected: pulmonary AFB+ TB, negative *Mtb* culture after 2 months of anti-TB treatment, appropriate follow-up during TB management to ensure culture conversion before 2 months of treatment without relapse (i.e. sampling after 15 days and 1 month of treatment), and complete follow-up 2 years after treatment completion.

**Bacterial strains.**   Mtb strains used in this study were isolated from pulmonary specimens during routine care in the Lyon University Hospital, France. The IMV and the 4MBE variant were obtained by cloning single-colony variants of the P1/L2 clinical isolate (Table 1). Briefly, bacterial suspension of the last isolate of P1/L2 patient was homogenized by vortexing in the presence of glass beads and Tween 80, and was plated on 7H10 supplemented with OADC to obtain single colonies. Clones were submitted to WGS to confirm the absence (IMV) or presence (4MBE) of the polymorphism of interest (C 3280555 G) and to confirm the absence of other mutations within the limits of the interpretation of short-read sequencing. Strains are stored at the Lyon University Hospital BSL3 laboratory and are available for research use if requested.

### TB-associated severity indices

**The Bandim TB score.**   The modified Bandim TB score considers 5 symptoms (cough, hemoptysis, dyspnea, chest pain, night sweats) and 5 clinical findings (anemia, tachycardia, positive finding at lung auscultation, fever, BMI [<18 and <16]), with one point for each. One clinical finding was excluded, the mid upper arm circumference as this data was not available in the Lyon University Hospital. Accordingly, patients were stratified into two severity classes, mild (Bandim score ≤4) and moderate or severe (≥5) [45–47].

**The MUST.**   The nutritional status of TB patients was evaluated thanks to the Malnutrition Universal Screening Tool (MUST), which includes three variables: unintentional weight loss score (weight loss < 5% = 0, weight loss 5–10% = 1, weight loss > 10% = 2), BMI score (BMI > 20.0 = 0, BMI 18.5–20.0 = 1, BMI < 18.5 = 2) and anorexia (if yes = 2). A MUST ≥ 4 is associated with poor TB prognosis [48].

**MIC determination.** Minimum inhibitory concentrations (MIC) were determined for RIF and INH (Sigma-Aldrich, Saint Louis, MO, USA) by a standard microdilution method as previously described [49]. Briefly, the assay was performed in a 96-well microtiter plate, with the concentrations ranging from 8 to 0.0008 mg/L, with a final inoculum of approximately $1.10^5$ CFU/mL in each well. The plates were incubated at 37°C for 12 to 18 days, and the wells were assessed for visible turbidity. The lowest concentration at which there was no visible turbidity was defined as the MIC.

**Antibiotics exposure using the BACTEC 960 system.** Antibiotic exposure experiments were performed in Mycobacterial Growth Indicator Tube (MGIT) using the BACTEC 960 system (Becton Dickinson, Sparks, MD, USA). Antibiotic solutions were added to MGIT at the required concentration and inoculated with $10^5$ CFU/mL for RIF exposure at 1 x MIC and with $2.10^6$ CFU/mL for RIF exposure at 4 x MIC. For the drug-free growth control, MGIT were inoculated with $10^3$ CFU/mL or $2.10^4$ CFU/mL, respectively. Analysis of the fluorescence was used to determine whether the tube was instrument positive; i.e. the test sample contains viable organisms, and results were expressed using MGIT time to positivity (TTP) system, reflecting bacterial growth [23,24]. The whole bacterial cultures were recovered to proceed to DNA or RNA extraction and purification.

**Droplet digital PCR.** ddPCR procedure was conducted using QX100 Droplet Digital PCR System (Bio-Rad Laboratories, Hercules, CA, USA). Total reaction volume was 22 μL, containing 11 μL of 2X ddPCR Supermix for Probes (No dUTPs), 8.9 μL of template DNA, 10 U of *Hind*III restriction enzyme and 1.1 μL of a 20X primers-probes mix (ddPCR mutant assay) provided by Bio-Rad (concentrations and sequences of probes are not provided by the manufacturer but they guaranty the validation by Minimum Information for Publication of Quantitative Real-Time PCR Experiments [MIQE] guidelines). The sequence studied was CGACACCGTTCGTGACCTCATCGCCCGTTGGGAGCAGCGGGACGTGATGGCGCG CGAGGTG[G/C]CCGTCGACGTGGCGTCGCACTCGCCTCAAGTCGATCCGATACTC GACGATTTGGCCGCGGC and the product was 70 bp. A no-template control and a positive control were used in each ddPCR batch. The procedure was carried out with droplet generation then PCR amplification with the following conditions: 95°C for 10 min followed by 40 cycles of 94°C for 30 s and 55°C for 1 min, and 98°C for 10 min, all steps with a ramp rate of 2°C/s. Analysis were done with QuantaSoft Analysis Pro software version 1.0.596 (Bio-Rad Laboratories). Positive droplets were counted and percentages of wild-type and mutant templates were calculated (% mutant = [(n mixed population/2)+n mutant]/total n amplified).

**DNA extraction, targeted NGS and WGS.** Genomic DNA was purified from cleared lysate, obtained after lysozyme and proteinase K digestion, using a QIAamp DNA mini Kit (Qiagen). For targeted NGS, we performed PCR using Platinum SuperFi DNA polymerase (Life Technologies SAS, Carlsbad, CA, USA) and primers indicated in S3 Table. DNA libraries were prepared with Nextera XT kit (Illumina, San Diego, CA, USA). Samples were sequenced on NextSeq or MiSeq system (Illumina) to produce 150 or 300 base-pair paired-end reads at the Bio-Genet NGS facility of Lyon University Hospital, as previously described [50].

**Bioinformatic analysis of Illumina data.** Reads were mapped with BOWTIE2 to the Mtb H37Rv reference genome (Genbank NC000962.2) and variant calling was made with SAMtools mpileup, as previously described [50]. For WGS, a valid nucleotide variant was called if the position was covered by a depth of at least 10 reads and supported by a minimum threshold rate of 10%. Regions with repetitive or similar sequences were excluded, i.e. PE, PPE, PKS, PPS, ESX. The WGS reference genome coverage ranges 96.5% to 99.0%, with an average depth of coverage of 46x to 185x. Regarding targeted NGS, a valid variant was called if the position was covered by a depth of at least 20 reads and supported by a minimum threshold rate of 0.1%. Depth of coverage of position of interest ranged from 10000 to 57000x.

Sequences have been submitted to European Nucleotide Archive (ENA) under accession number PRJEB37306.

**Apolar lipid extraction and analysis by gas chromatography–mass spectrometry (GC-MS).** Mycobacteria were grown in Middlebrooks 7H9 medium supplemented with 10% OADC (oleic acid, albumin, dextrose, catalase) until mid-log phase and pelleted before being washed twice in sterile water. The pellet was resuspended in methanol: 0.3% aqueous NaCl (10:1) and petroleum ether was added to separate the liquid layers. The non-aqueous petroleum ether extracts containing apolar lipids were dried under nitrogen. Apolar lipids were hydrolyzed with KOH 5% in methanol at 100˚C for 90 min. The reaction mixture was acidified with 3N HCl and acidic water was added. Fatty acids released were extracted with chloroform-ethanol (3:1). The upper phase was removed and lower phase stored. The extracted fatty acids were derivatized with pentafluorobenzyl bromide (PFB) and analyzed in the negative ion mode by gas chromatography-mass spectrometry (GC-MS/MS Agilent HP7890B/7000C). The gas chromatograph was performed at the Functional Lipidomics platform, acknowledged by Infrastructure in Biology, Health and Agronomy (IBiSA).

**Killing kinetics of RIF and INH alone and combination and mathematical modeling.** *In vitro* experiments of the antibacterial effect of INH and RIF alone and in combination, as well as the mathematical modeling of the results were performed as described in a previous publication [49].

*Killing kinetics of single drug.* For each strain, an inoculum of approximately $5.10^5$ CFU/mL was prepared in 7H9 Middlebrook medium. The inoculum was incubated without any antibiotic (growth control) and with RIF, at concentrations of 1/32, 1/16, 1/8, 1/4, 1/2, 1, 2, 4, 16, 64, and 256 multiples of MIC (xMIC). Incubation was performed in 96 deepwell plates (1 ml working volume) at 37˚C. Each well was homogenized by pipetting and vortexing, serially diluted, and plated on 7H10 agar. The numbers of CFU per milliliter was determined at day 7 of drug exposure. Each experiment was performed twice.

For drug tolerance experiments, an inoculum of approximately $2.10^7$ CFU/mL was prepared in 7H9 Middlebrook medium. The inoculum was incubated without any antibiotic (growth control) or with RIF, at concentrations of 2, 4, 8, and 16xMIC. At days 3, 5, 7, and 10 of drug exposure bacteria were processed as described above for CFU counting to determine subsequently the $MDK_{99}$ of each RIF doses [51,52]. Each experiment was performed thrice.

*Killing kinetics of INH + RIF combination.* For each strain, a bacterial inoculum was prepared as described above. Twelve concentrations of IHN (0, 1/32, 1/16, 1/8, 1/4, 1/2, 1, 2, 4, 8, 16, and 64 multiples of MIC) were combined with 12 possible concentrations of RIF (same multiples of MIC as INH) in an incomplete checkerboard design. The incubation was performed as for single drug experiments. The numbers of CFU per milliliter was determined after day 7 of drug exposure. A total of 142 bacterial counts were available for each strain.

*Mathematical modelling of single drug effect.* A Hill pharmacodynamic equation was used to describe the antibacterial effect of single drugs [53]. The model was fit to single drug log-transformed data, as follows:

$$E = \frac{E_{max}}{1 + 10^{\log_{10}\left[\left(\frac{EC_{50}}{C}\right)^H\right]}} = \frac{E_{max}}{1 + 10^{(H \times (\log_{10}(EC_{50}) - \log C))}} \quad (1)$$

Where E is the drug effect, C is the drug concentration, $E_{max}$ is the maximal effect $EC_{50}$ is the median effect concentration, and H is the Hill coefficient of sigmoidicity. GraphPad Prism for Windows version 5.02 (GraphPad Software, La Jolla, CA, USA) was used for model fitting and parameter estimation.

The antibacterial effect E considered was the net difference between the bacterial count measured with no drug (C = 0) and the count for a given drug concentration at the same time-point. Drug concentrations were normalized to the MIC, so C is expressed as the ratio of the actual concentration of drug in culture divided by the MIC.

*Response-surface modeling of INH and RIF combined effect.* We used the Minto model [49,54] to describe the combined antimicrobial effect of INH and RIF based on data measured after 7 days of therapy.

The Minto model is an extension of the Hill equation to drug combination and is defined as follows as follows:

$$E = \frac{E_{max}(U) \cdot \left(\frac{X_{INH}+X_{RIF}}{U_{50}(U)}\right)^{H(U)}}{1 + \left(\frac{X_{INH}+X_{RIF}}{U_{50}(U)}\right)^{H(U)}} \tag{2}$$

With $X_{INH} = \frac{C_{INH}}{EC_{50,INH}}$ and $X_{RIF} = \frac{C_{RIF}}{EC_{50,RIF}}$

And $U = \frac{X_{RIF}}{X_{INH}+X_{RIF}}$

In those equations, $X_{INH}$ and $X_{RIF}$ are concentrations of INH and RIF normalized to the drug potency, U quantifies the ratio of each drug in the combination, $E_{max}(U)$ is the maximal effect at ratio U, $U_{50}(U)$ is the number of units of $X_{RIF}$ associated with 50% of the maximal effect at ratio U, and H(U) is the coefficient of sigmoidicity at ratio U. Each of the three parameters of the Hill equation are function of U, the drug ratio. The concentration-effect curve associated with each ratio defined the contour of a response-surface of the drug combination.

As suggested by Minto et al., we used a polynomial function to describe $E_{max}(U)$, $U_{50}(U)$, and H(U), as described below:

$$E_{max}(U) = E_{max,INH} + (E_{max,RIF} - E_{max,INH} - \beta_{Emax}) \cdot U + \beta_{Emax} \cdot U^2 \tag{3}$$

where $E_{max,INH}$ and $E_{max,RIF}$ are the maximal effect of INH and RIF alone, respectively, and $\beta_{Emax}$ is the coefficient of the two-order polynomial for $E_{max}(U)$;

$$H(U) = H_{INH} + (H_{RIF} - H_{INH} - \beta_H) \cdot U + \beta_H \cdot U^2 \tag{4}$$

where $H_{INH}$ and $H_{RIF}$ are the Hill coefficient of sigmoidicity for INH and RIF alone, respectively, and $\beta_H$ is the coefficient of the two-order polynomial for H(U); and

$$U_{50}(U) = 1 - \beta_{U50} \cdot U + \beta_{U50} \cdot U^2 \tag{5}$$

Where $\beta_{U50}$ is the coefficient of the two-order polynomial for $U_{50}(U)$.

The last equation allows to interpret the response surface in terms of synergy/antagonism. If $\beta_{U50}$ is not different from 0, $U_{50}$ is equal to 1 for all values of U, and the interaction is additive. If $\beta_{U50}$ is significantly greater than 0, the curve shows an inward curvature and the normalized drug mixture, is more than additive, which means synergy. If $\beta_{U50}$ is significantly lower than 0, the curve displays an outward curvature, and the normalized drug mixture ($X_{INH}$ + $X_{RIF}$)/$U_{50}(U)$) is less than additive, which means antagonism.

The Minto model was fit to data by using non-linear regression within the Matlab software (version 2011b; The Mathworks, Natick, MA, USA). Point estimates of model parameters along with their confidence intervals were obtained. Goodness-of-fit of the model was assessed by analysis of plots of observed versus predicted antibacterial effects.

Values and 95% confidence intervals of the $\beta_{U50}$ coefficient were examined to interpret the combined action in terms of synergy/antagonism. Synergy was confirmed when the lower

bound of the 95% confidence interval was greater than 0. Antagonism was stated when the upper bound of the 95% confidence interval was lower than 0. When the 95% confidence interval included 0, the interaction was considered as additive.

**Cell Culture and Infection protocol.** U937 cells, monocytic cell line from pleural effusion, cultured in RPMI-1640 medium supplemented with 10% fetal bovine serum were differentiated with phorbol 12-myristate 13-acetate (PMA; Sigma-Aldrich) 100nM for 48h at a density of $2.10^5$ cells/mL. Adherent cells were then infected with bacterial suspension at a multiplicity of infection (MOI) of 10:1 (bacteria to cells) in antibiotic-free culture medium, at 37˚C in 5% $CO_2$ atmosphere, as previously described [55].

**Intracellular CFU counting.** U937 cells differentiated cells were infected by Mtb, as described above. After 5 h of culture, infected cells were incubated with amikacin 200 mg/L for 1 h to remove the extracellular Mtb. The cells were washed thrice to remove amikacin. To study Mtb uptake by macrophages, cells were immediately lysed with distilled water containing saponin 0.1% for 10 min and intracellular mycobacteria were plated on complete 7H10 agar and incubated for 3–4 weeks. To measure intra-cellular multiplication, separate wells with amikacin-treated inoculated cells were further incubated for 96 h at 37˚C in 5% $CO_2$ atmosphere. After that, extracellular bacteria were removed by 1 h of incubation with amikacin 200 mg/L and intracellular bacteria were counted as describe above.

**Immunofluorescence and image acquisition.** U937 differentiated cells were infected by Mtb, as described above, before staining for confocal microscopy analysis, 6 h and 24 h post-infection (hpi), as previously described [55]. For the study of autophagic flux, 3 h before staining, cells were incubated with chloroquine (Sigma-Aldrich) 40μM to avoid acidification of autophagy compartments. Infected cells were washed thrice and incubated with LysoTracker Red DND-99 500nM (Thermo Fisher Scientific Inc, Rockford, IL, US; L7528) for 1 h to detect acidify compartments. Following 4% paraformaldehyde fixation over 20 min, cells were incubated with 100 mM glycine for 20 min and washed with PBS. Cells were then permeabilized with 0.1% Triton X-100 for 10 min and saturated in PBS-3% bovine serum albumin (BSA) for 30 min. Cells were labelled with anti-Mtb coupled FITC antibody (Abcam, Cambridge, MA, US; ab20962, 1/100 dilution) and anti-LC3B coupled Dy-Light 650 antibody (to assess autophagy activation; Thermo Fisher Scientific, PA5-22937, 1/400 dilution) in PBS-3% BSA-0.05% Triton X100 for 1h at room temperature. Slides were mounted in Fluoromount medium (Sigma -Aldrich) and observed with a Leica confocal microscope SP5X. Images were acquired with Leica Application Suite software. Stacks of confocal images and quantification of percent of colocalization were performed with ImageJ software [56] and JACoP plugin.

**CCF-4 assay for translocation of Mtb in host cell cytosol.** To detect mycobacterial escape from the phagosome and translocation in host cell cytosol, the CCF4 FRET assay was performed [28,57]. Briefly, infected cells were stained with 50 μM CCF4 (Invitrogen, Carlsbad, CA, USA) in buffer containing anion transport inhibitor (Invitrogen) for 2 h at room temperature, as recommended by the manufacturer. Cells were washed with PBS containing anion transport inhibitor before fixing with paraformaldehyde (PFA, Sigma-Aldrich) 4% for 30 min at room temperature in the dark. Cells were washed before performing directly fluorescence acquisition. Intact CCF4-AM emits green fluorescence (520 nm) due to FRET between the fluorescent moieties, indicating that mycobacteria are in intracellular compartments. Cleavage of CCF4-AM, due to β-lactamase expressed by cytosolic Mtb, leads to a shift in the fluorescence emitted to bleu (450nm). The results obtained were normalized on intracellular bacterial load.

**Measurement of lactate dehydrogenase (LDH) in cell culture supernatants.** U937 cells differentiated cells were infected by Mtb, as described above. To quantify necrosis of U937 cells, the LDH release in cell culture supernatant from infected cells was measured at 24 h and

96 hpi by using the colorimetric kit (Roche, Mannheim, Germany) and following the manufacturer's instructions.

**Quantitative determination of apoptosis.** U937 differentiated cells were infected by Mtb, as described above, before staining for cell death analysis, at 24 hpi and 96 hpi. Infected cells were washed thrice and apoptosis was detected by Annexin V FITC (Thermo Fisher Scientific) after 15min of incubation at room temperature. After fixing with PFA 4% for 30min, cells were analyzed by using fluorescent microscopy.

**Infected macrophage RNA extraction.** U937 differentiated cells were infected by Mtb, as described above, before RNA extraction. Total macrophage RNA was obtained from Mtb-infected cells at 24 hpi using the Trizol method as previously described [58]. Briefly, macrophages were lysed with Trizol reagent (Qiagen) and total RNA was isolated with miRNeasy mini kit according to the manufacturer's instruction (Qiagen). The macrophage RNA was subjected to DNaseI digestion before final purification.

**Transcriptomic analysis.** Transcriptomic analysis was performed at ViroScan3D/ProfileXpert platform (www.viroscan3d.com). Briefly, RNA from each sample was quantified using Quantus HSRNA method (Promega) and qualified using Fragment analyzer HSRNA (AATI). After quality control, total RNA were submitted to polyA capture using NextFlex poly(A) Beads (PerkinElmer, Waltham, MA, USA), then mRNA were submitted to library preparation using NextFlex Rapid Directional mRNA (PerkinElmer) and 100 ng input. As recommended by the ENCODE (Encyclopedia of DNA Elements) consortium ERCC (External RNA Control Consortium) RNA spike-In (Invitrogen) were added to samples in order to ensure reproducibility of the experiments. Single-read sequencing with 75 bp read length was performed on NextSeq 500 high output flowcell (Illumina). Mapped reads for each samples were counted and normalized using FPKM method [59]. Fold change between the different groups were calculated using median of groups and p-value of difference were calculated using t-test with equal variance without p-value correction in the RStudio v.0.99.893 (RStudio Inc., Boston, MA, USA). Heatmaps were performed thanks to Heatmapper [60], using average linkage clustering method and Pearson distance measurement method.

**Cytokine production assay.** U937 differentiated cells were infected by Mtb, as described above, and cell supernatant was recovered 6 hpi, 24 hpi and 96 hpi. Cell culture supernatants were screened for the presence of 27 human cytokines and chemokines using the Bio-Plex Pro Human Cytokine Standard 27-Plex kit (Bio-Rad Laboratory) on a FLEXMAP 3D analyzer (Luminex, Austin, TX, USA). Data were analyzed using Bio-Plex Manager software version 6.1 (Bio-Rad Laboratory).

**Statistical analysis.** Statistical analyses were performed using GraphPad Prism for Windows, version 5.02 (GraphPad Software, La Jolla, CA, USA). Data obtained from lipidomic analysis, RIF exposure experiments, variant frequency analysis by ddPCR, intracellular CFU counting, lytic cell death, as well as cytokine and chemokine production were expressed as mean ± standard deviation (SD). Means were compared using One-Way ANOVA or Repeated Measures ANOVA followed with Bonferroni correction, unpaired t-test or Student's t-test, where appropriate. For data obtained from pharmacological models, parameter values were given as point estimate [95% confidence interval, CI]. For confocal microscopy analysis and macrophage apoptosis, data were expressed as median values [interquartile range, IQR]. Results were compared by using Kruskal-Wallis analysis, using Dunn's Multiple Comparison Test or Mann Whitney test, where appropriate. $^*p<0.05$, $^{**}p<0.01$, $^{***}p<0.001$.

## Supporting information

**S1 Fig. Fitness of isolates from patients with delayed culture conversion upon RIF exposure.** Mycobacterial growth of the initial isolate (white), last isolate (grey), and last isolate after

1xMIC RIF exposure (black) from patients with delayed culture conversion, in absence (Growth control, inoculated with $2.10^4$ CFU/mL) or presence of RIF (black; 4x minimum inhibitory concentration (MIC) RIF, inoculated with $2.10^6$ CFU/mL) is expressed in time to positivity (TTP) in the BACTEC system. Symbols represent values of two independent experiments. Means were compared using Repeated Measures ANOVA followed by Bonferroni correction.
(TIF)

**S2 Fig. Time-kill curves of IMV and 4MBE under RIF exposure at various MICs.** Mycobacterial growth of the IMV (black) and 4MBE variant (white) expressed as the percentage of survival obtained thanks to CFU count, without rifampicin (RIF) exposure (A, growth control), at 2xMIC (B), 4xMIC (C), 8xMIC (D) and 16xMIC (E) of RIF. The symbols represent the mean of three independent experiments; the bars represent standard deviation.
(TIF)

**S3 Fig. Improved Mtb intra-macrophagic survival of the 4MBE variant.** At 24 hpi, cells were stained with anti-Mtb coupled FITC. (A) The proportion of infected cells were determined across at least 10 confocal images, containing between 30 and 80 cells per image, per replicate. (B) Mtb staining area was determined across at least 40 cells per replicate. Values for each condition are the median values [interquartile range, IQR] of at least three independent experiments. Statistical significance was determined using Mann Whitney test. $^*p<0.05$, $^{**}p<0.01$, $^{***}p<0.001$.
(TIF)

**S4 Fig. Early inflammatory response is slightly exacerbated upon infection by the 4MBE variant.** (A) TNF-α, (B) IL-6, (C) IL-1β, (D) IFN-γ, (E) CCL5 or RANTES and (F) Interferon gamma-induced protein 10 (IP-10) release in cell culture supernatant was evaluated at 6 hpi. Values for each condition are the mean ± standard deviation (SD) of three independent experiments. Means were compared using Repeated Measures ANOVA followed by Bonferroni correction. $^*p<0.05$, $^{**}p<0.01$, $^{***}p<0.001$.
(TIF)

**S5 Fig. Persistent inflammatory response upon infection by the 4MBE variant.** (A) TNF-α, (B) IL-6, (C) IL-1β, (D) IFN-γ, (E) CCL5 or RANTES and (F) Interferon gamma-induced protein 10 (IP-10) release in cell culture supernatant was evaluated at 96 hpi. Values for each condition are the mean ± standard deviation (SD) of three independent experiments. Means were compared using Repeated Measures ANOVA followed by Bonferroni correction. $^*p<0.05$, $^{**}p<0.01$, $^{***}p<0.001$.
(TIF)

**S1 Table. Model parameters and fit for rifampicin (RIF) effect measured after 7 days.**
(DOCX)

**S2 Table. Parameter values and goodness of fit of the response-surface model describing the combined action of INH and RIF measured after 7 days.**
(DOCX)

**S3 Table. Primer sets for targeted NGS.**
(DOCX)

## Acknowledgments

The authors thank Philip Robinson and Véréna Landel (DRCI, Hospices Civils de Lyon) for help in manuscript preparation.

## Author Contributions

**Conceptualization:** Charlotte Genestet, Sylvain Goutelle, Oana Dumitrescu.

**Data curation:** Charlotte Genestet, Elisabeth Hodille, Jean-Luc Berland, Sylvain Goutelle, Oana Dumitrescu.

**Formal analysis:** Charlotte Genestet, Elisabeth Hodille, Alexia Barbry, Jean-Luc Berland, Jonathan Hoffmann, Fabiola Bastian, Michel Guichardant, Samuel Venner, Christophe Ginevra, Sylvain Goutelle, Oana Dumitrescu.

**Funding acquisition:** Charlotte Genestet, Jean-Luc Berland, Samuel Venner, Gérard Lina, Sylvain Goutelle, Oana Dumitrescu.

**Investigation:** Charlotte Genestet, Elisabeth Hodille, Alexia Barbry, Jonathan Hoffmann, Fabiola Bastian, Michel Guichardant.

**Methodology:** Charlotte Genestet, Christophe Ginevra, Sylvain Goutelle, Oana Dumitrescu.

**Project administration:** Charlotte Genestet, Sylvain Goutelle, Oana Dumitrescu.

**Resources:** Sylvain Goutelle, Oana Dumitrescu.

**Software:** Charlotte Genestet, Jean-Luc Berland, Emilie Westeel.

**Supervision:** Charlotte Genestet, Sylvain Goutelle, Oana Dumitrescu.

**Validation:** Charlotte Genestet, Elisabeth Hodille, Alexia Barbry, Jonathan Hoffmann, Fabiola Bastian, Michel Guichardant.

**Visualization:** Charlotte Genestet, Elisabeth Hodille, Sylvain Goutelle, Oana Dumitrescu.

**Writing – original draft:** Charlotte Genestet, Sylvain Goutelle, Oana Dumitrescu.

**Writing – review & editing:** Charlotte Genestet, Elisabeth Hodille, Alexia Barbry, Jean-Luc Berland, Jonathan Hoffmann, Emilie Westeel, Fabiola Bastian, Michel Guichardant, Samuel Venner, Gérard Lina, Christophe Ginevra, Florence Ader, Sylvain Goutelle, Oana Dumitrescu.

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
