## [Decision Letter · Decision Letter 0]

6 Jan 2021

Dear Dr. Genestet,

Thank you very much for submitting your manuscript "Rifampicin exposure reveals within-host Mycobacterium tuberculosis diversity in patients with delayed culture conversion" for consideration at PLOS Pathogens. As with all papers reviewed by the journal, your manuscript was reviewed by members of the editorial board and by several independent reviewers. In light of the reviews (below this email), we would like to invite the resubmission of a significantly-revised version that takes into account the reviewers' comments.

The reviewers raised important points that require additional work. The authors should particularly focus on performing the drug tolerance experiments to get a true readout of tolerance as well as including isolates from other patients in the same cohort as well as isolates from a control cohort. Thus, the drug tolerance experiments need to be done properly so as to get a more accurate readout of tolerance. When repeating this work, it would be advisable to include isolates from other patients in the delayed culture conversion cohort and control isolates from patients in a cohort without delayed culture conversion. Sequencing of the isolates from the patients without delayed culture conversion would provide the necessary sequencing data to ascertain how common the reported minor genetic variation really is.

We cannot make any decision about publication until we have seen the revised manuscript and your response to the reviewers' comments. Your revised manuscript is also likely to be sent to reviewers for further evaluation.

Sincerely,

Helena Ingrid Boshoff

Associate Editor

PLOS Pathogens

JoAnne Flynn

Section Editor

PLOS Pathogens

Kasturi Haldar

Editor-in-Chief

PLOS Pathogens

orcid.org/0000-0001-5065-158X

Michael Malim

Editor-in-Chief

PLOS Pathogens

orcid.org/0000-0002-7699-2064

The reviewers raised important points that require additional work. The authors should particularly focus on performing the drug tolerance experiments to get a true readout of tolerance as well as including isolates from other patients in the same cohort as well as isolates from a control cohort. Thus, the drug tolerance experiments need to be done properly so as to get a more accurate readout of tolerance. When repeating this work, it would be advisable to include isolates from other patients in the delayed culture conversion cohort and control isolates from patients in a cohort without delayed culture conversion. Sequencing of the isolates from the patients without delayed culture conversion would provide the necessary sequencing data to ascertain how common the reported minor genetic variation really is.

Reviewer's Responses to Questions

**Part I - Summary**

Reviewer #1: The study aims to provide insights in the evolution of M. tuberculosis in the host with a particular focus on delayed treatment response, resistance development, and antibiotic tolerant sub-populations. The used a combination of microbiological analysis with WGS to unravel mutant populations in the initial cultures that may be enriched during therapy.

I am a bit uncertain about the experimental procedures. On particular point is that the antibiotic tolerant phenotype has not genetic trait, so it is reversible. How can this then be captured?

Also, is there any evidence that the pre-treatment with 1 x MIC of RIF is the right dosage for pre-selection?

While the paper is well written overall, however, several points need to be addressed in a revised version. This is esp. the case for the selection and macrophage experiments. How many times have they been done? It there any statistics done demonstrating that this is not chance?

The variant detected is quite interesting, if this is really providing selective advantage in case of RIF treatment, the variant should be seen in clinical isolates already. This should be checked as genome data are available from a high number of clinical isolates.

Reviewer #2: The authors run a screening on several isolates from patients with delayed culture conversion, sequencing both the first and the last isolates coming from those patients. They also grow the last isolates from the patients in the presence of an inhibitory dose of rifampicin and sequence the resulting bacterial population. They find a series of variant mutations and decide to study one patient in particular. They manage to isolate single colony variants from this patient, one carrying one potentially interesting mutation (4MBE variant) and one identical to the initial majority variant in the patient (IMV). They demonstrate that the 4MBE variant has selective advantage over the IMV in the presence of rifampicin even if they have the same MIC and that the 4MBE variant has increased ability to infect macrophages and replicate or survive intramacrophagically, and the variant increases apoptosis and inflammatory response in macrophages.

The authors are aiming to establish a link between micro-diversity in clinical isolates and delayed culture conversion by means of low-frequency variants with increased tolerance for the rifampicin and increased persistence in macrophages. They study one low frequency variant from a patient with delayed culture conversion and convincingly demonstrate that the variant has selective advantage both in the presence of rifampicin and in the infection of macrophages (in vitro). However, I do not think that the main claims of the paper are supported by the evidence as presented. This is not because their conclusions are necessarily incorrect per se, but rather because their results are insufficient to support those conclusions. Although the authors claim to be studying within-host evolution, most of the paper is focused on a single patient with no clear rationale for doing so and the data from the rest of the cohort are very basic and not explored further. In my opinion, they should either re-frame the paper or find additional evidence.

Reviewer #3: Overall I think this is a worthwhile manuscript. It covers a topic of current interest, namely the role of unfixed genetic variants in the outcomes of tuberculosis disease. Strengths are the fact that it includes both clinical and genetic data, and then explores functional consequences of findings, which is often not done by equivalent papers.

A weakness of the clinical aspect is that no sequencing from control sequences are provided - it is therefore difficult to ascertain using the authors' sequencing/variant calling pipeline how common minor genetic variation is. Also, the paper would flow better if the authors were able to explain why the focussed on the mas minor variant, as currently the narrative jumps from the genetic section to the molecular biology section.

**Part II – Major Issues: Key Experiments Required for Acceptance**

Reviewer #1: Line 77 - 80 – please include references for the statements

Results

Line 122 – it is important to detail the experimental procedures – what was used for DNA extraction for the culture, how deep was sequenced? Also, how were the minor variants been identified? Any particular statistical method for that?

Line 123 – how was the RIF pretreatment done? The overall procedure is not really clear to me, but hast a strong influence of the outcome. How many repetitions did you do? Any statistics done?

Line 137 – please explain a bit more precisely what you did – how was the single colony plating done? Was it from the initial culture or the subsequent one?

Line 139 - targeted NGS of what? How was this done?

Line 144 – the macrophage experiments are not clear to me. How long are they performed? How many generations of mycobacterial growth would you expect? Please provide growth curves etc. Even if experiments are done for one week – the number of generations for potential selection of the variant is really low. So, please provide more data on it. Also, did you do the experiments in repetition to so that this does not happen by chance? Also, did you check the inoculum and/or the pre-culture for the frequencies?

Line 149 – what do you mean with “cloned variants”.

Line 147 to 153 – I did not understand what was done here, please clarify

Line 166 – what do you mean with significantly here – did you do statistics? If yes, include values.

Line 181 – difficult to believe that if the MIC is not changed, the growth in 1x MIC is better for the mutant. Indeed, now growth should be observed if the strains are treated with 1 x MIC of RIF - or? So, either the MIC finding is wrong – or the time to culture positivity data make no sense.

Also, general drug tolerance experiments are done differently. You need to expose a culture with high dose treatment and then do CFU over time to measure the treatment refractory proportion of the culture. This should be done for the mutant clone and the parental strain to see if there is really a higher drug tolerance level.

Line 219 – do you mean uptake?

Discussion

General: In host diversity has been described before, so I do find the data presented here add little further knowledge. However, the 4MBE variant is really interesting. If the overall advantages of the variant are as drastic as described, it should already be found in nature. As all the experiments performed in vitro, it would be great if the authors can screen WGS data collections for the presence of the variant in clinical isolates. The tolerance phenotype is not really clear to me, as the authors did not really perform drug tolerance experiments. It would also be great to get an idea about the mechanism.

Reviewer #2: - The authors do not sequence or present any data from patients without delayed culture conversion. If they are arguing that the micro-diversity found in those patients is relevant we would need a control cohort for comparison. We would need a rifampicin exposure experiment with those control isolates to show that samples from delayed culture conversion patients have more new variants than the controls.

- The authors find several mutations that appear only after exposure to rifampicin in several patients, but they do not go back and try to find if these mutations were in the patients before the exposure using targeted sequencing. Furthermore, the authors do not show any evidence that these mutations (particularly the one in the variant they focus on) are not present in patients with early culture conversion or sequence isolates from any such patients for comparison. Also, if the mutation in 4MBE is responsible for the late culture conversion of the patient and has (limited) selective advantage in the presence of rifampicin and confers some advantage in macrophage infection, why is it such a minority variant in the isolates from the patient?

- It is unclear to me why the authors decide to focus on a particular patient and not any of the others. They should explain better why they chose to focus on this patient specifically. If there is no clear explanation, they should test isolates from other patients as well for tolerance to rifampicin and/or performance in macrophage infection.

- The authors say that 4MBE has increased tolerance, but I do not think they are measuring tolerance. Tolerance is the ability to withstand the antibiotic even at high concentrations and it is generally measured using killing curves in which a high dose of the antibiotic is applied and the rate at which bacteria die is measured. Instead, the authors are measuring a mix of survival and growth ability even at sub-MIC concentrations. This kind of measurement is closer to relative fitness than to tolerance, in my opinion. I think that this shows that 4MBE has selective advantage but not tolerance, the authors should drop the term or show data that demonstrate said increase in tolerance. For their killing kinetics the authors say that they are taking measurements after 3, 5, 7 and 10 days of rifampicin exposure but they show only the results corresponding to day 7. Also, some of the rifampicin concentrations they are using are well above the MIC . This means that they could use those data to actually determine bacterial death rates and thus check for differences in tolerance. Why didn’t the authors do that?

Reviewer #3: For the minor variant work, as a control of their variant calling pipeline, how many minor variants did they see in control sequences (e.g. baseline samples from patients who did culture-convert rapidly).

**Part III – Minor Issues: Editorial and Data Presentation Modifications**

Reviewer #1: (No Response)

Reviewer #2: - The authors focus on intramacrophagic survival/replication of the 4MBE variant but give no data on overall replication or rationale about why that is not relevant.

Reviewer #3: lines 38-40: I am not quite sure how these correspond to the samples listed in table 1. For the 9/12 patients with unfixed SNPs - which patients are these and which timepoint is referred to? Same for the 6/12 with nsSNPs.

lines 80: rephrase mandatory - strongly recommended, but some places still don't!

'as it is known to hamper treatment efficacy' - reword to make it clear 'it' is drug resistance, not drug resistance detection

Table 1: How often were cultures performed? Were cultures positive on a monthly basis until the time indicated? For those patients that died, after how many months' treatment did they die?

Lines 135: was there a reason for focusing on this strain?

Line 302: as above, not sure what these numbers refer to

Lines 430-438: how was bioinformatic analysis of targeted sequencing performed? What coverage depth was achieved and what variant calling thresholds were used?

PLOS authors have the option to publish the peer review history of their article (what does this mean?). If published, this will include your full peer review and any attached files.

Reviewer #1: No

Reviewer #2: No

Reviewer #3: No
---

## [Editor Report · Decision Letter 1]

13 May 2021

Dear Dr. Genestet,

We are pleased to inform you that your manuscript 'Rifampicin exposure reveals within-host Mycobacterium tuberculosis diversity in patients with delayed culture conversion' has been provisionally accepted for publication in PLOS Pathogens.

Best regards,

Helena Ingrid Boshoff

Associate Editor

PLOS Pathogens

JoAnne Flynn

Section Editor

PLOS Pathogens

Kasturi Haldar

Editor-in-Chief

PLOS Pathogens

orcid.org/0000-0001-5065-158X

Michael Malim

Editor-in-Chief

PLOS Pathogens

orcid.org/0000-0002-7699-2064

The authors have addressed all the major concerns.
---

## [Editor Report · Acceptance letter]

4 Jun 2021

Dear Dr. Genestet,

We are delighted to inform you that your manuscript, "Rifampicin exposure reveals within-host Mycobacterium tuberculosis diversity in patients with delayed culture conversion," has been formally accepted for publication in PLOS Pathogens.

Best regards,

Kasturi Haldar

Editor-in-Chief

PLOS Pathogens

orcid.org/0000-0001-5065-158X

Michael Malim

Editor-in-Chief

PLOS Pathogens

orcid.org/0000-0002-7699-2064